# PrismAudio: Decomposed Chain-of-Thoughts and Multi-dimensional Rewards for Video-to-Audio Generation

**Huadai Liu**[1,2]**, Kaicheng Luo**[2]**, Wen Wang**[2]**, Qian Chen**[2]
**Peiwen Sun**[3]**, Rongjie Huang**[3]**, Xiangang Li**[2]**, Jieping Ye**[2]**, Wei Xue**[1]*
[1]Hong Kong University of Science and Technology (HKUST)
[2]Tongyi Fun Team, Alibaba Group
[3]The Chinese University of Hong Kong (CUHK)

## Abstract

Video-to-Audio (V2A) generation requires balancing four critical perceptual dimensions: *semantic consistency*, *audio-visual temporal synchrony*, *aesthetic quality*, and *spatial accuracy*; yet existing methods suffer from objective entanglement that conflates competing goals in single loss functions and lack human preference alignment. We introduce **PrismAudio**, the first framework to integrate Reinforcement Learning into V2A generation with specialized Chain-of-Thought (CoT) planning. Our approach decomposes monolithic reasoning into four specialized CoT modules (Semantic, Temporal, Aesthetic, and Spatial CoT), each paired with targeted reward functions. This CoT-reward correspondence enables **multidimensional RL optimization** that guides the model to *jointly* generate better reasoning across all perspectives, solving the objective entanglement problem while preserving interpretability. To make this optimization computationally practical, we propose **Fast-GRPO**, which employs hybrid ODE-SDE sampling that dramatically reduces the training overhead compared to existing GRPO implementations. We also introduce **AudioCanvas**, a rigorous benchmark that is more distributionally balanced and covers more realistically diverse and challenging scenarios than existing datasets, with 300 single-event classes and 501 multi-event samples. Experimental results demonstrate that PrismAudio achieves state-of-the-art performance across all four perceptual dimensions on both the in-domain VGGSound test set and out-of-domain AudioCanvas benchmark. The project page is available at `https://PrismAudio.github.io`.

## 1 Introduction

Video-to-Audio (V2A) Generation, also known as video foley, aims to synthesize a soundscape from a silent video and an optional text input. Achieving satisfactory V2A results is not merely about generating plausible acoustics; the generated audio needs to meet criteria across four distinct human perceptual axes: (a) **semantic consistency**, ensuring audio events correspond accurately to visual content, (b) **temporal synchrony**, aligning audio timing precisely with visual cues, (c) **aesthetic quality**, capturing the subjective richness, complexity, and artistic value that makes audio perceptually satisfying and creatively useful, (d) **spatial accuracy**, measuring the accuracy of the left-right sound image w.r.t. traditional stereo. Mastering these axes is crucial for enabling genuine controllability—the ability to articulate not just *what* to render but *how*—freeing creators from the constraints of opaque, end-to-end models. Yet, this multi-objective challenge proves overwhelming for current methods: semantic and temporal alignment are brittle in complex scenes; aesthetic quality is subjective and hard to quantify; spatial accuracy remains underexplored; and the objectives are inherently interdependent and have a trade-off relationship. For example, a system focusing solely on semantic consistency may generate a mundane sound with low aesthetic quality, or generate the right type of sound but fail on temporal synchronization. Unable to navigate this complex landscape

---
*Corresponding author.

of competing goals, models regress to optimizing for signal-level reconstruction, fundamentally failing to bridge the gap between model outputs and true human perceptual expectations.

Recent V2A advances (Zhang et al., 2024; Xing et al., 2024; Wang et al., 2024b) have evolved from direct synthesis to increasingly rich conditioning mechanisms. Early approaches like V2A-Mapper (Wang et al., 2024a) and Diff-Foley (Luo et al., 2023) rely solely on visual inputs, using embedding projection and contrastive alignment, respectively, but suffer from limited semantic precision and controllability. Subsequent methods (Chen et al., 2025; Mo et al., 2024; Tian et al., 2025) incorporate explicit text conditioning—MovieGen Audio (Polyak et al., 2024) via cross-attention in diffusion transformers, MMAudio (Cheng et al., 2024a) through multimodal transformers—yet remain opaque "black boxes" despite improved control. Most recently, ThinkSound (Liu et al., 2025b) pioneers Chain-of-Thought (CoT) reasoning (Wei et al., 2022) using multimodal LLMs (MLLMs) (Achiam et al., 2023; Cheng et al., 2024b; Chu et al., 2024), decomposing V2A into structured planning followed by audio rendering. This explicit reasoning significantly enhances interpretability and narrative coherence by making the generation process transparent and controllable.

However, ThinkSound still exhibits three critical limitations: First, its *monolithic planning* generates all audio analysis through a single reasoning path, conflating distinct analytical tasks—semantic understanding, synchronization, spatial reasoning, and aesthetic evaluation—and leading to inadequate treatment of each dimension and multimodal hallucinations in complex scenarios. Second, *objective entanglement* forces the model to optimize a unified reconstruction loss that conflates competing perceptual goals—narrative coherence, temporal synchrony, aesthetic quality, and spatial accuracy—without learning appropriate context-dependent trade-offs, particularly in complex scenarios demanding sophisticated multi-objective reasoning. Third, *the absence of human preference alignment* means the model lacks mechanisms to learn perceptually satisfying audio beyond textual matching, producing technically correct but perceptually unsatisfying results. **While the first limitation is specific to ThinkSound, the latter two afflict all existing V2A approaches**.

To address these limitations, we introduce **PrismAudio**, the first framework to tightly integrate Reinforcement Learning (RL) into V2A generation with specialized CoT planning. We decompose ThinkSound's monolithic planning into four specialized CoT modules—**Semantic CoT**, **Temporal CoT**, **Aesthetic CoT**, and **Spatial CoT**—each providing focused, interpretable reasoning for its corresponding perceptual dimension. Crucially, we pair each CoT module with targeted reward signals. The CoT-reward correspondence enables **multidimensional RL optimization** that guides all modules to **jointly generate better reasoning across all perspectives**, fundamentally addressing objective entanglement and lack of human preference alignment while preserving interpretability.

PrismAudio builds upon a CoT-aware audio foundation model employing a Multimodal Diffusion Transformer backbone with flow matching. Applying RL to diffusion models poses computational challenges (Xue et al., 2025; Li et al., 2025). While Group Relative Policy Optimization (GRPO) (Shao et al., 2024) shows promise for human preference alignment, current implementations like Flow-GRPO (Liu et al., 2025c) require Stochastic Differential Equation (SDE) sampling at every denoising step, creating substantial training overhead due to full-step sampling requirements for policy ratio computation. We propose **Fast-GRPO**, employing a hybrid ODE-SDE strategy—applying SDE sampling only to a subset of steps for stochastic exploration while using deterministic Ordinary Differential Equation (ODE) sampling elsewhere. Fast-GRPO enables efficient multi-dimensional CoT-RL optimization without compromising generation quality.

Evaluating *practical* V2A capabilities demands a rigorous benchmark covering realistically diverse and challenging scenarios; yet existing V2A benchmarks such as VGGSound (Chen et al., 2020) and Kling-Audio-Eval (Wang et al., 2025) fail to meet the requirements (see Appendix C.2 for detailed analysis). We therefore introduce **AudioCanvas**, featuring: (1) **high modality alignment** through rigorous off-screen sound filtering, (2) **advanced scene complexity** with 300 single-event classes and 501 multi-event samples across diverse scenes, and (3) **precise audio captions with rich, structured CoT reasoning** enabling comprehensive evaluation of semantic consistency, temporal synchrony, aesthetic quality, and spatial accuracy. Our main contributions are as follows:

- We introduce **PrismAudio**, the first V2A framework to integrate specialized CoT modules with multi-dimensional RL optimization, fundamentally addressing limitations of existing approaches.
- We propose **Fast-GRPO**, enabling efficient multi-dimensional RL training of diffusion models through hybrid ODE-SDE sampling.

- We construct **AudioCanvas**, a rigorous V2A benchmark spanning diverse scenes with strict quality control and high-quality annotations, providing challenging real-world V2A evaluations.
- Extensive experiments demonstrate that PrismAudio outperforms baselines across all perceptual axes on both the VGGSound test set and AudioCanvas. Further analysis reveals that single-dimensional rewards suffer from suboptimal trade-offs—improving one dimension at others' expense—while our multi-dimensional RL optimization framework balances all objectives without compromising individual performance.

## 2 RELATED WORK

**CoT Reasoning for Audio Generation.** Large Language Models (LLMs) (Guo et al., 2025; Team et al., 2024b; Yang et al., 2025) have demonstrated remarkable reasoning capabilities through CoT prompting (Wei et al., 2022), enabling complex problem decomposition via intermediate reasoning steps. This paradigm has been extended to MLLMs, which integrate visual and audio understanding with linguistic reasoning (Achiam et al., 2023; Lin et al., 2023; Alayrac et al., 2022). The related works on V2A generation are summarized in Appendix A. Early V2A approach (Xie et al., 2024) uses vision-language models for video captioning, then employs text-to-audio models for synthesis. Recent works adopt video-audio-language MLLMs like VideoLLaMA2 (Cheng et al., 2024b) for structured CoT planning. ThinkSound (Liu et al., 2025b) exemplifies this by generating detailed audio descriptions before synthesis, improving semantic consistency and narrative coherence. However, existing MLLM-based approaches employ monolithic planning that cannot handle competing objectives or provide targeted optimization for distinct perceptual dimensions. Our work decomposes monolithic planning into four specialized CoT modules—*Semantic*, *Temporal*, *Aesthetic*, and *Spatial*—each providing focused reasoning with corresponding reward signals for multi-dimensional preference optimization.

**Reinforcement Learning for Diffusion Models.** Reinforcement Learning has achieved remarkable success in LLMs through RLHF (Ouyang et al., 2022; Bai et al., 2022), demonstrating the crucial role of aligning model outputs with human preference beyond likelihood maximization. Recent works have explored RL applications to diffusion models for preference alignment. Early approaches (Fan & Lee, 2023; Black et al., 2023; Fan et al., 2023) optimize diffusion score functions through policy gradient methods, while Wallace et al. (2024) introduces DPO (Rafailov et al., 2023) to diffusion models for direct learning from human feedback. Most recently, Group Relative Policy Optimization (GRPO) (Shao et al., 2024) based approaches have advanced RL-enhanced diffusion models. Flow-GRPO (Liu et al., 2025c) and DanceGRPO (Xue et al., 2025) introduce GRPO to flow matching models (Lipman et al., 2022), enabling divergent sampling by transforming ODEs into equivalent SDEs with reduced variance through group-based optimization. However, existing RL approaches for generation primarily focus on single-objective optimization and have not been extended to V2A generation, which expects multi-dimensional alignment across semantic, temporal, aesthetic, and spatial aspects. Our work pioneers the application of flow-matching GRPO to V2A generation with specialized multi-dimensional reward decomposition.

## 3 PRISMAUDIO

As illustrated in Figure 1, our method consists of three main stages built on an audio foundation model. Section 3.1 presents the CoT-aware audio foundation model. Section 3.2 elaborates the customized CoT modules that decompose V2A reasoning into four specialized dimensions: *Semantic*, *Temporal*, *Aesthetic*, and *Spatial*, where each module generates targeted reasoning text that provides dimension-specific guidance for audio generation. Finally, Section 3.3 introduces our GRPO post-training framework, which includes multi-dimensional reward design that aligns with our specialized CoT modules, and our Fast-GRPO algorithm that enables efficient multi-objective optimization across all perceptual dimensions.

### 3.1 CoT-AWARE AUDIO FOUNDATION MODEL

We build our audio foundation model on the diffusion transformer backbone (Peebles & Xie, 2023; Liu et al., 2023b) with flow matching that takes video inputs and text conditioning to generate audio outputs. It undergoes standard pre-training on large-scale video-audio pairs to establish basic generation capabilities. While this architecture provides a solid foundation for V2A generation,

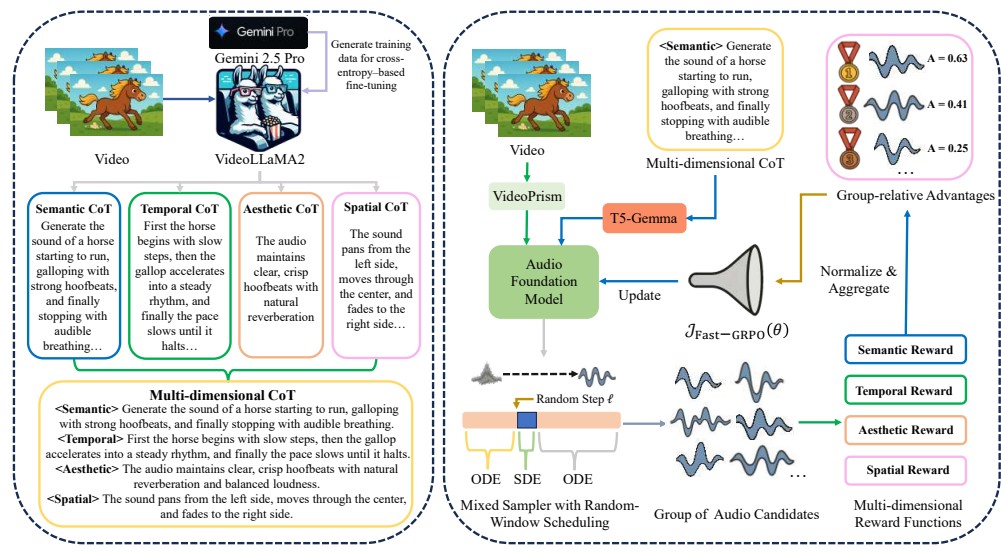

Figure 1: Overview of **PrismAudio**. Left panel: the progress of CoT training data construction using Gemini 2.5 Pro and then fine-tuning VideoLLaMA2 for decomposed CoT generation (Section 3.2). Right panel: the Fast-GRPO multi-dimensional CoT-RL framework (Section 3.3) for post-training the Audio Foundation Model (Section 3.1).

it has two critical limitations that hinder effective multi-dimensional CoT reasoning: *insufficient video understanding for complex and diverse scenarios*, and *limited text processing capabilities for structured reasoning content*. Therefore, we enhance the ThinkSound architecture (Liu et al., 2025b) with the following two modifications to facilitate multi-dimensional CoT reasoning.

**VideoPrism for Enhanced Video Understanding.** Most existing V2A models, including ThinkSound, adopt CLIP-based image encoders (Radford et al., 2021) that process video frames *independently as static images*. This approach lacks comprehensive video understanding and fails to handle complex, diverse video scenarios in real-world applications. We replace CLIP with Video-Prism (Zhao et al., 2024), a state-of-the-art (SOTA) video encoder pre-trained on large-scale video data. VideoPrism employs a unified vision transformer architecture specially designed for video understanding, capturing rich semantic representations of objects, actions, and environmental contexts that are crucial for our multi-dimensional reasoning modules.

**T5-Gemma for CoT-Aware Text Encoding.** Our CoT modules produce analytical text containing logical structures, causal relationships, and multi-faceted reasoning patterns that require sophisticated language understanding; yet the standard T5 encoders in ThinkSound struggle with the complex, structured reasoning text generated by our CoT modules. Hence, we make another essential enhancement by upgrading the T5 encoder to T5-Gemma (Zhang et al., 2025). T5-Gemma adapts the reasoning capabilities of decoder-only LLMs into an efficient encoder-decoder architecture, effectively enabling proper conditioning on our structured CoTs for the generation model with its stronger reasoning comprehension capabilities.

## 3.2 DECOMPOSING MULTI-DIMENSIONAL COT REASONING

While ThinkSound proves the effectiveness of CoT reasoning for V2A generation, it generates all audio-related analysis in a *single, undifferentiable* reasoning path. This monolithic reasoning has critical limitations: different aspects of audio generation require fundamentally different analytical frameworks—semantic understanding focuses on content identification, spatial reasoning requires directional positioning logic, and aesthetic evaluation demands subjective quality assessment. When these diverse tasks are conflated, models struggle to properly address each dimension and often introduce multimodal hallucinations when reasoning about multiple complex aspects simultaneously.

To achieve superior reasoning capabilities simultaneously across all four dimensions, we first employ Gemini 2.5 Pro (Comanici et al., 2025) for CoT data construction, leveraging its outstanding multimodal understanding and strong reasoning capabilities. Next, using this high-quality training data, we fine-tune the highly competitive open-source video language model VideoLLaMA2 (Cheng

et al., 2024b) to generate four specialized CoTs: **Semantic CoT** identifies audio events and their characteristics from audio-visual content; **Temporal CoT** determines the sequential ordering of audio events; **Aesthetic CoT** focuses on audio quality aspects like naturalness and fidelity; and **Spatial CoT** analyzes sound positioning, including directional placement and distance for proper spatialization. The four specialized CoTs are then concatenated in this order to form the *multi-dimensional CoT* (as depicted in Figure 1) and used as *enhanced, structured text conditioning* to fine-tune our audio foundation model (Section 3.1), enabling the model to learn from explicit reasoning patterns and acquire better generalization by understanding the underlying logic behind audio-visual correspondences.

### 3.3 THE FAST-GRPO MULTI-DIMENSIONAL RL FRAMEWORK

#### 3.3.1 MULTI-DIMENSIONAL REWARD FUNCTIONS

As explained in Section 1, V2A generation involves multiple human perceptual objectives that are inherently interdependent and conflicting. High-quality audio requires simultaneous success across semantic accuracy, temporal coherence, aesthetic quality, and spatial positioning—objectives that often compete with each other. A monolithic reward function struggles to balance these competing objectives and often leads to suboptimal trade-offs where improvements in one dimension come at the expense of others, as illustrated in Table 4.

To address this limitation, we design four specialized reward functions that align with our CoT dimensions: **Semantic Reward**, measured by MS-CLAP (Elizalde et al., 2024), a commonly used audio-text alignment model for evaluating content similarity; **Temporal Reward**, assessed via Synchformer (Iashin et al., 2024), a highly competitive model specifically designed to detect audio-visual synchrony; **Aesthetic Reward**, which uses the leading-edge assessor for audio aesthetic quality, Meta Audiobox Aesthetics (Tjandra et al., 2025), as a no-reference model trained to predict human Mean Opinion Scores (MOS); and **Spatial Reward**, which employs the high-performing StereoCRW (Chen et al., 2022) to verify directional positioning accuracy. This multi-objective approach allows for a balanced and comprehensive optimization across all key perceptual dimensions.

#### 3.3.2 FAST-GRPO WITH RANDOM-WINDOW EXPLORATION

To align the audio foundation model with the multi-dimensional human preference, we adopt GRPO for its stability. While generation of our flow matching model is inherently deterministic (an ODE), it can be equivalently formulated as a stochastic process (an SDE) that enables RL-based optimization (see Appendix B.1 for details). Prior works (Xue et al., 2025; Liu et al., 2025c) construct the Markov Decision Process (MDP) within the GRPO training process by applying this SDE formulation across the *entire* denoising trajectory. This "pure SDE" approach, however, forces GRPO to evaluate the policy at every step, creating a significant efficiency bottleneck.

To resolve this trade-off between exploration and efficiency, we introduce **Fast-GRPO**. Its core idea is to **strategically confine stochasticity and optimization to a small, computationally inexpensive segment of the generation process.** We achieve this by creating a hybrid sampling path: an efficient, deterministic ODE is used for most of the trajectory, while an explorative SDE is activated only within a small, randomly placed window of timesteps. Fast-GRPO is realized through two key components: a mixed ODE–SDE sampler and a random-window scheduling scheme.

**Mixed Sampler with Random-Window Scheduling.** For each training iteration, we randomly sample a starting position $\ell \in \{0, 1, \ldots, T - w\}$. This defines an optimization window $\mathcal{W}(\ell)$ with width $w \ll T$:

$$\mathcal{W}(\ell) = \{\ell, \ell + 1, \ldots, \ell + w - 1\}. \tag{1}$$

We then interleave deterministic ODE steps and stochastic SDE steps based on this window. For a step size $\Delta t$, the update rule is:

$$\mathbf{x}_{t+1} = \begin{cases} \mathbf{x}_t + v_\theta(\mathbf{x}_t, t, c)\Delta t, & \text{if } t \notin \mathcal{W}(\ell) \quad \text{(ODE step)} \\ \mathbf{x}_t + \mu_{\text{SDE}}(\mathbf{x}_t, t, c)\Delta t + \sigma_t\sqrt{\Delta t}\,\varepsilon_t, & \text{if } t \in \mathcal{W}(\ell) \quad \text{(SDE step)} \end{cases} \tag{2}$$

where $\varepsilon_t \sim \mathcal{N}(0, I)$, $v_\theta$ is the model's predicted velocity, and $\mu_{\text{SDE}}$ is the SDE drift term derived from $v_\theta$ (see Appendix B.1). This hybrid approach is theoretically sound, as it preserves the terminal data distribution required for correct reward computation (see Appendix B.2).

**Per-step policy and ratio.** The SDE steps within the window $\mathcal{W}(\ell)$ induce a tractable Gaussian policy $\pi_\theta(\mathbf{x}_{t+1} \mid \mathbf{x}_t, c)$, allowing for a closed-form computation of the GRPO policy ratio $r_t(\theta)$ (see Appendix B.3 for derivation). This policy and its corresponding GRPO ratio are given by:

$$\pi_\theta(\mathbf{x}_{t+1} \mid \mathbf{x}_t, c) = \mathcal{N}\Big(\mu_\theta(\mathbf{x}_t, t, c), (\sigma_t^2 \Delta t)I\Big), \tag{3}$$

$$r_t(\theta) = \exp\Big\{ - \frac{\|\mathbf{x}_{t+1} - \mu_\theta\|_2^2 - \|\mathbf{x}_{t+1} - \mu_{\theta_{\text{old}}}\|_2^2}{2\sigma_t^2 \Delta t} \Big\}, \tag{4}$$

where $\mu_\theta(\mathbf{x}_t, t, c) = \mathbf{x}_t + \mu_{\text{SDE}}(\mathbf{x}_t, t, c)\Delta t$.

**Multi-reward, Group-relative Advantages.** Given $K$ reward heads $\{R_k\}_{k=1}^K$ that are aligned with our CoT dimensions (Semantic, Temporal, Aesthetic, and Spatial), we sample a group of $N$ audio candidates $\{\mathbf{x}_T^i\}_{i=1}^N$ per prompt $c$ with the old policy. We first compute a weighted total reward for each candidate:

$$R_{\text{total}}^i = \sum_{k=1}^K \lambda_k R_k(\mathbf{x}_T^i, c). \tag{5}$$

The advantage score $A^i$ is then computed by normalizing this aggregated reward using the group's mean ($\mu_{\text{group}}$) and standard deviation ($\sigma_{\text{group}}$):

$$A^i = \frac{R_{\text{total}}^i - \mu_{\text{group}}}{\sigma_{\text{group}} + \epsilon}, \qquad \text{where } \mu_{\text{group}} = \frac{1}{N} \sum_{j=1}^N R_{\text{total}}^j \text{ and } \sigma_{\text{group}} = \text{std}\big(\{R_{\text{total}}^j\}_{j=1}^N\big). \tag{6}$$

A small constant $\epsilon$ (e.g., $10^{-6}$) is added to the denominator for numerical stability. This approach preserves GRPO's stability through within-group normalization while enabling principled multi-objective trade-offs via the weights $\lambda_k$.

**Windowed GRPO Objective.** The policy model is optimized by maximizing the following objective, derived from the Fast-GRPO formulation restricted to the selected SDE steps:

$$\mathcal{J}_{\text{Fast-GRPO}}(\theta) = \mathbb{E}_{c,\ell,\{\mathbf{x}^i\}\sim\pi_{\theta_{\text{old}}}} \left[ \frac{1}{N} \sum_{i=1}^N \frac{1}{w} \sum_{t\in\mathcal{W}(\ell)} \min\Big(r_t^i(\theta)\, A^i, \; \text{clip}(r_t^i(\theta), 1-\varepsilon, 1+\varepsilon)\, A^i\Big) \right]. \tag{7}$$

where $A^i$ is the group-normalized advantage for the $i$-th sample (Eq. 6). This design reduces the policy-model NFE (Number of Function Evaluations) from $T$ to $w$ per sample, yielding a near-linear complexity of GRPO training. We notice that some **contemporaneous** works, such as Mix-GRPO (Li et al., 2025), also propose hybrid ODE-SDE. Considering the concurrency of research and their differences from Fast-GRPO on window design, modalities, and scopes, our Fast-GRPO is a valid innovation for enabling efficient multi-dimensional RL training of diffusion models.

## 4 EXPERIMENTS

### 4.1 EXPERIMENTAL SETUP

**AudioCanvas Benchmark.** To address critical gaps in V2A evaluation—lack of scene complexity and high-quality, structured annotations—we introduce **AudioCanvas**, a new benchmark of 3,177 real-world videos. It is uniquely distinguished by three core features: (1) **High-Fidelity Alignment**, ensured through rigorous, expert-led manual filtering, addressing known quality issues in existing datasets; (2) **Advanced Scene Complexity**, featuring the first curated set of 501 multi-event scenarios to test performance beyond simple events; and (3) **Rich, Structured Annotations**, with CoT reasoning generated by Gemini 2.5 Pro and quantitatively validated to over 94% human-verified accuracy. Appendix C details the construction, quality assessment, and benchmark comparisons.

**Evaluation Metrics.** We conduct comprehensive evaluations using both *objective* and *subjective* metrics to assess the four key perceptual dimensions. For objective evaluation, we adopt established metrics across multiple dimensions. Following ThinkSound, we employ **CLAP** score for text-audio semantic alignment, **DeSync** measured by Synchformer for video-audio temporal synchrony, Fréchet Distance (**FD**) (Kilgour et al., 2018) in the VGGish feature space, and Kullback-Leibler (**KL**) Divergence (Copet et al., 2024) based on predictions from the PaSST model for audio distribution similarity Liu et al. (2025b). For spatial accuracy of generated stereo audio, we adopt

Table 1: **Objective and Subjective evaluations** on the **in-domain** VGGSound test set. Best results are in **bold**. *PrismAudio w/o CoT-RL* is our audio foundation model without the multi-dimensional CoT conditioning and Fast-GRPO post-training. We report the mean and standard deviation of the MOS scores. We evaluate all the open-sourced baselines except for those with [†], which denote evaluation using generation samples released by the authors. **Time(s)** denotes the inference time (excluding feature extraction) for generating 9-second audio samples.

| Method | Params | Semantic CLAP↑ | Temporal DeSync↓ | Aesthetic Quality | | | | Spatial Accuracy | | Distribution | | Subjective | | Time (s) |
|---|---|---|---|---|---|---|---|---|---|---|---|---|---|---|
| | | | | PQ↑ | PC↓ | CE↑ | CU↑ | GCC↓ | CRW↓ | FD↓ | KL↓ | MOS-Q↑ | MOS-C↑ | |
| GT | - | 0.46 | 0.55 | 6.30 | 3.85 | 4.40 | 5.65 | - | - | - | - | 4.58±0.18 | 4.65±0.15 | - |
| Frieren[†] | 159M | 0.32 | 0.85 | 5.90 | 3.50 | 3.57 | 5.35 | - | - | 1.34 | 2.86 | 3.45±0.75 | 3.51±0.80 | - |
| V2A-Mapper[†] | 229M | 0.31 | 1.23 | 6.26 | 3.54 | 4.12 | 5.63 | - | - | 0.90 | 2.49 | 3.38±0.82 | 3.44±0.88 | - |
| AudioX | 1.1B | 0.41 | 1.24 | 5.94 | 3.43 | 3.86 | 5.44 | 7.22 | 19.25 | 1.51 | 1.80 | 3.61±0.75 | 3.65±0.72 | 7.52 |
| HunyuanVideo-Foley | 5.31B | 0.42 | 0.55 | 5.85 | 3.26 | 3.92 | 5.26 | - | - | 2.26 | 1.73 | 3.88±0.55 | 3.96±0.52 | 10.63 |
| MMAudio | 1.03B | 0.40 | 0.46 | 5.94 | 3.51 | 3.88 | 5.28 | - | - | 2.17 | 1.32 | 3.95±0.51 | 4.03±0.58 | 1.30 |
| ThinkSound | 1.3B | 0.43 | 0.55 | 6.15 | 3.53 | 3.95 | 5.48 | 4.65 | 13.47 | 1.17 | 1.35 | 4.05±0.55 | 4.18±0.51 | 1.07 |
| **PrismAudio (Ours)** | 518M | **0.47** | **0.41** | **6.38** | 3.24 | 4.29 | 5.68 | **3.77** | **7.72** | 1.08 | 1.23 | **4.21±0.35** | **4.22±0.29** | 0.63 |
| PrismAudio w/o CoT-RL | 518M | 0.42 | 0.51 | 6.17 | 3.32 | 3.94 | 5.48 | 4.06 | 10.29 | 1.14 | 1.43 | 4.02±0.48 | 4.11±0.42 | 0.63 |

**GCC** MSE and **CRW** MSE Sun et al. (2024) to evaluate both difference of arrival (DoA) and interaural time difference (ITD). To measure aesthetic quality, we evaluate production quality (**PQ**), production complexity (**PC**), content enjoyment (**CE**), and content usefulness (**CU**) scores from Audiobox-Aesthetics (Tjandra et al., 2025). For subjective evaluation, we employ **Mean Opinion Score (MOS)** across two complementary dimensions: **MOS-Q (Quality)** evaluates the aesthetic quality and audio fidelity of generated audio, while **MOS-C (Consistency)** evaluates the comprehensive alignment between audio and video, encompassing semantic consistency, temporal synchrony, and spatial accuracy. More details of evaluation metrics are in Appendix F.

**Implementation Details** are in Appendix D. Since the multi-dimensional CoT fine-tuning and the RL post-training are based on VGGSound, evaluations on the VGGSound test set are **in-domain** evaluations. Competitive **baselines** include Frieren (16k, mono) (Wang et al., 2024b), V2A-Mapper (16k, mono) (Wang et al., 2024a), AudioX (44k, stereo) (Tian et al., 2025), HunyuanVideo-Foley (44k, mono) (Shan et al., 2025), MMAudio (44k, mono) (Cheng et al., 2024a), and ThinkSound (44k, stereo) (Liu et al., 2025b).

## 4.2 Main Results

**In-domain Evaluation on VGGSound Test Set.** We compare our PrismAudio against competitive open-source V2A baselines on the VGGSound test set, with results shown in Table 1. We observe that: (1) **PrismAudio achieves new SOTA performance across all perceptual dimensions.** Compared to the prior SOTA, ThinkSound, our model shows substantial gains in semantics (CLAP: **0.47** vs. 0.43) and synchrony (DeSync: **0.41** vs. 0.55), while slashing the spatial CRW error from 13.47 to **7.72**. Subjective evaluations corroborate these gains, with PrismAudio achieving the highest MOS scores for both quality and content consistency. (2) **Our CoT-RL framework is the key driver of performance gains.** Our ablation model, *PrismAudio w/o CoT-RL*, already constitutes an impressively strong baseline that outperforms prior SOTA models in multiple metrics (e.g., DeSync, CRW). The CoT-RL optimization then provides a substantial further boost **across all dimensions**, including **4.7%** and **2.7%** relative gains on MOS-Q and MOS-C. This clearly demonstrates that by decomposing CoT reasoning and applying targeted rewards, our approach effectively resolves objective conflicts and substantially improves the performance of a highly optimized foundation model. (3) **PrismAudio is also more efficient.** With much fewer parameters than prior SOTAs, it achieves superior performance with faster inference, making it far more practical for real-world applications.

**Out-of-Domain Evaluation on AudioCanvas.** To assess generalizability, we evaluate models on our challenging AudioCanvas benchmark. The results in Table 2 can conclude that: (1) **PrismAudio demonstrates exceptional robustness while other models falter.** On this complex data, most baselines suffer significant degradation; the prior SOTA, ThinkSound, collapses in temporal reasoning (DeSync: 0.80) and spatial accuracy (CRW: 22.82). In contrast, PrismAudio remains stable, achieving the best MOS scores and even surpassing the ground truth in semantic alignment and synchrony.[1] These results prove that PrismAudio learns true audio-visual principles, not just overfitting. (2) **The benefit of our CoT-RL framework is amplified on complex data.** The framework's contribution is even more critical here than on VGGSound, substantially elevating performance over the

---

[1]These remarkable results occur because our RL framework is powerful enough to explicitly optimize for the target metrics. While ground truth audio contains natural variations that these imperfect proxies may penalize, our model can generate audio that better meets the criteria of the metrics. Crucially, our high MOS scores demonstrate that this enhanced control also results in superior perceptual quality for human listeners.

ablation model across all dimensions (e.g., semantics: CLAP: 0.47 → **0.52**; aesthetics: CE: 3.81 → **4.26**). This widening performance gap confirms our multi-dimensional CoT-RL framework is indispensable when simple pattern matching fails. Detailed breakdown results are in Appendix E.4.

Table 2: **Objective and Subjective evaluations** on the **out-of-domain** AudioCanvas benchmark.

| Method | Semantic CLAP↑ | Temporal DeSync↓ | Aesthetic Quality | | | | Spatial Accuracy | | Distribution | | Subjective | |
|---|---|---|---|---|---|---|---|---|---|---|---|---|
| | | | PQ↑ | PC↓ | CE↑ | CU↑ | GCC↓ | CRW↓ | FD↓ | KL↓ | MOS-Q↑ | MOS-C↑ |
| GT | 0.48 | 0.40 | 6.47 | 3.16 | 4.02 | 5.99 | - | - | - | - | 4.65±0.23 | 4.72±0.20 |
| HunyuanVideo-Foley | 0.44 | 0.47 | 6.43 | 3.25 | 4.04 | 5.88 | - | - | 2.04 | 2.07 | 3.75±0.52 | 3.71±0.58 |
| MMAudio | 0.46 | 0.43 | 6.30 | 3.23 | 3.97 | 5.77 | - | - | 3.59 | 1.87 | 3.88±0.45 | 3.87±0.41 |
| ThinkSound | 0.48 | 0.80 | 6.48 | 3.50 | 4.10 | 5.94 | 4.43 | 22.82 | 1.95 | 2.54 | 3.79±0.58 | 3.80±0.54 |
| **PrismAudio (Ours)** | **0.52** | **0.36** | **6.68** | **2.82** | **4.26** | **6.15** | 3.50 | 12.87 | 1.92 | 1.53 | **4.12±0.28** | **4.01±0.25** |
| PrismAudio (Silent Video CoT) | 0.47 | 0.42 | 6.55 | 3.04 | 4.09 | 5.98 | 3.63 | 14.26 | 2.01 | 1.79 | - | - |
| PrismAudio w/o CoT-RL | 0.42 | 0.44 | 6.45 | 3.22 | 3.81 | 5.87 | 4.11 | 15.30 | 2.10 | 2.17 | 3.91±0.35 | 3.85±0.31 |

## 4.3 ABLATION AND ANALYSIS

We conduct comprehensive ablation studies and analysis to evaluate critical algorithmic designs and provide deeper insights. Multi-dimension CoT Reasoning and RL analyses on **VGGSound test set** are in Appendix E.2. For the audio foundation model, analyses of its video encoder and text encoder are in Appendix E.1, and more analyses about it are in Appendix E.3.

**Multi-dimensional CoT Reasoning.** To validate the design principles of our multi-dimensional CoT, we analyze several reasoning strategies on AudioCanvas, with results presented in Table 3. Our analysis yields two primary findings: (1) **Structured reasoning is essential for high-quality generation.** The necessity of CoT reasoning is immediately evident when comparing against the *Baseline (No CoT)*, which performs poorly across all metrics, with particularly weak semantic alignment (CLAP: 0.42) and spatial accuracy (CRW: 15.30). Furthermore, not just any reasoning suffices. The *Random CoT* variant—which contains the correct concepts but in a jumbled, illogical structure—improves upon the baseline but fails to match coherent CoTs. Its poor aesthetic (CE) and spatial scores prove that a structured, logical plan is vital, not merely a bag of conceptual keywords. (2) **Decomposed reasoning is superior to a monolithic approach.** This is the core advantage of our framework. Our *MultiCoT* substantially outperforms the *Monolithic CoT* (ThinkSound-style), especially in semantic understanding (CLAP: **0.52** vs. 0.46) and aesthetic quality (CE: **4.26** vs. 3.79). These results strongly support our hypothesis that a single, conflated reasoning block struggles to balance competing objectives, leading to inter-dimensional interference.

**Training Efficiency of Fast-GRPO.** We compare our Fast-GRPO against Flow-GRPO, which employs SDE sampling across the entire trajectory. Figure 2 illustrates the training curves of both methods, tracking the *Semantic* reward score over the training steps. The results reveal the substantial advantages of our method: (1) **Fast-GRPO exhibits drastically faster convergence and higher training efficiency**. It surpasses the final performance of Flow-GRPO (∼0.47) in just 200 steps, while Flow-GRPO requires more than 600 steps to reach its plateau. (2) **Fast-GRPO also achieves a considerably higher final reward score** [2], reaching ∼0.51 compared to Flow-GRPO's 0.47, indicating that our hybrid ODE-SDE approach not only improves training efficiency but also leads to a better optimization outcome.

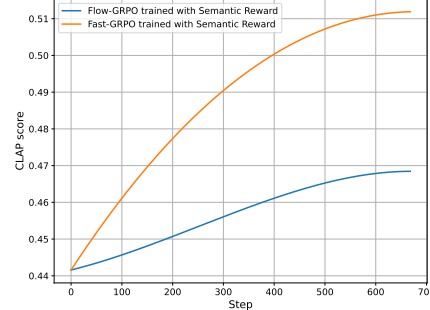

Figure 2: Training convergence on **Semantic** reward measured by the CLAP score.

**Multi-dimensional vs. Single-dimensional Rewards.** To demonstrate the necessity of holistic optimization, especially on complex out-of-domain data, we compare our multi-dimensional RL approach against single-dimensional alternatives. As shown in Table 4, we observe that: (1) **Single-dimensional optimization leads to severe objective entanglement.** While each specialized model excels at its target metric, it comes at a great cost to others. For instance, the *Semantic Only* model achieves the highest CLAP score (0.54), but its temporal synchronization breaks down, with DeSync

---

[2] Generally higher GRPO reward scores could correspond to better final performance, but other factors may interfere with the correlation.

Table 3: Analysis of different CoT reasoning strategies on AudioCanvas. *MultiCoT* denotes decomposed, multi-block reasoning in our PrismAudio, *Monolithic CoT* denotes unified, single-block reasoning as in ThinkSound, and *Random CoT* denotes structurally corrupted monolithic reasoning.

| Method | Semantic CLAP↑ | Temporal DeSync↓ | Aesthetic Quality | | | | Spatial Accuracy | | Distribution | |
| | | | PQ↑ | PC↓ | CE↑ | CU↑ | GCC↓ | CRW↓ | FD↓ | KL↓ |
|---|---|---|---|---|---|---|---|---|---|---|
| Baseline (No CoT) | 0.42 | 0.44 | 6.45 | 3.22 | 3.81 | 5.87 | 4.11 | 15.30 | 2.10 | 2.17 |
| Random CoT | 0.44 | 0.41 | 6.30 | 2.94 | 3.78 | 5.96 | 3.92 | 13.79 | 2.06 | 1.75 |
| Monolithic CoT | 0.46 | 0.38 | 6.34 | 2.89 | 3.79 | 5.99 | 3.92 | 13.02 | 1.96 | 1.70 |
| MultiCoT | **0.52** | **0.36** | **6.68** | **2.82** | **4.26** | **6.15** | **3.50** | **12.87** | **1.92** | **1.53** |

Table 4: Analysis of multi-dimensional vs. single-dimensional reward functions with our multi-dimensional CoTs on AudioCanvas.

| Reward Focus | Semantic CLAP↑ | Temporal DeSync↓ | Aesthetic Quality | | | | Spatial | | Distribution | |
| | | | PQ↑ | PC↓ | CE↑ | CU↑ | GCC↓ | CRW↓ | FD↓ | KL↓ |
|---|---|---|---|---|---|---|---|---|---|---|
| Baseline (No RL) | 0.47 | 0.42 | 6.45 | 3.02 | 3.81 | 5.87 | 4.11 | 15.30 | 1.90 | 1.58 |
| Semantic Only | **0.54** | 0.58 | 6.62 | 2.91 | 3.93 | 6.11 | 3.53 | 11.89 | 1.84 | **1.49** |
| Temporal Only | 0.46 | **0.35** | 6.39 | 3.05 | 3.63 | 5.71 | 4.29 | 13.08 | 1.88 | 1.68 |
| Aesthetic Only | 0.46 | 0.42 | **7.06** | **2.61** | **3.92** | **6.48** | 4.08 | 13.51 | 4.50 | 1.92 |
| Spatial Only | 0.47 | 0.42 | 6.44 | 3.01 | 3.72 | 5.80 | **3.16** | **11.88** | **1.77** | 1.67 |
| Multi-dimensional | 0.52 | 0.36 | 6.68 | 2.82 | 4.26 | 6.15 | 3.50 | 12.87 | 1.92 | 1.53 |

error increasing from 0.42 to 0.58. Most strikingly, the *Aesthetic Only* model, while reaching a super-high PQ of 7.06, more than doubles the distribution metric (FD) (from 1.90 to 4.50), indicating it generates audio that sounds "pleasing" in isolation but is semantically detached from the video's content and context. (2) **Our multi-dimensional rewards successfully balance these trade-offs.** In stark contrast, our approach is the only method that achieves balanced, holistic improvements. It simultaneously enhances all key aspects over the baseline: semantics (CLAP: $0.47 \rightarrow 0.52$), temporal synchrony (DeSync: $0.42 \rightarrow 0.36$), aesthetic quality (PQ: $6.45 \rightarrow 6.68$), and spatial accuracy (CRW error: $15.30 \rightarrow 12.87$). These results clearly demonstrate that as task complexity increases, concurrently optimizing across all perceptual axes becomes indispensable to avoid catastrophic failures and generate audio that is coherent, synchronized, and perceptually satisfying.

## 4.4 CASE STUDY

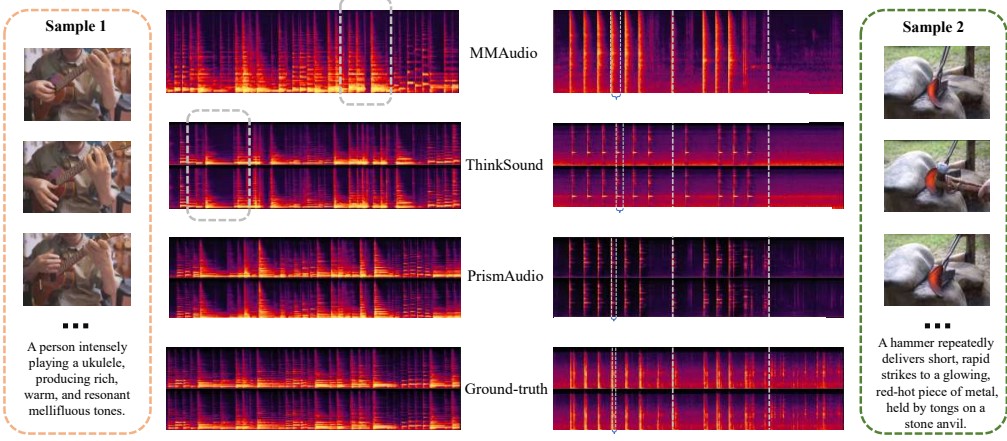

Figure 3: Qualitative comparison of PrismAudio against baseline models.

We present a qualitative analysis in Figure 3 and observe that: (1) Aesthetic Quality & Musical Fidelity: In the ukulele scene (left), PrismAudio achieves high musical fidelity, matching the ground-truth's clean harmonics and rich high-frequency details. In contrast, ThinkSound suffers from significant high-frequency loss (dashed box), while MMAudio produces a blurry, smeared mono spectrogram, demonstrating their failure to preserve aesthetic quality. (2) Transient Response & Temporal Synchrony: In the blacksmith scene (right), PrismAudio accurately renders sharp, high-energy transients (hammer strikes), maintaining temporal synchrony in line with the ground-truth.

ThinkSound's transients are noticeably weaker, and MMAudio exhibits severe temporal smearing and artifacts (dashed line), failing to align with the visual events.

## 5 CONCLUSION

We introduce PrismAudio, a novel framework that, for the first time, integrates multi-dimensional CoT reasoning with reinforcement learning for V2A generation. By decomposing monolithic planning into four specialized perceptual dimensions—Semantic, Temporal, Aesthetic, and Spatial—and aligning them with corresponding reward signals, our approach directly addresses objective entanglement and lack of human preference alignment that have limited prior works. Comprehensive experiments on existing benchmarks and our new, challenging AudioCanvas benchmark demonstrate that PrismAudio achieves SOTA performance by successfully balancing all competing objectives, establishing a new controllable and interpretable paradigm for V2A generation.

## REPRODUCIBILITY STATEMENT

To ensure reproducibility, the code, the AudioCanvas benchmark, and all model weights will be made publicly available upon publication. Core implementation details are provided in Appendix D. The released package will include:

- Complete training scripts and configuration files required to reproduce the main results.
- The training dataset generated by Gemini 2.5 Pro for the VideoLLaMA2 fine-tuning stage.
- Detailed documentation covering the experimental setup and hyperparameter settings.

## ACKNOWLEDGEMENT

The research was supported in part by Early Career Scheme (ECS-HKUST22201322), Theme-based Research Scheme under Grant T45-205/21-N from Hong Kong RGC, and Generative AI Research and Development Centre from InnoHK.

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

## A  RELATED WORK ON VIDEO-TO-AUDIO GENERATION

V2A generation has recently gained significant attention with advances in multimodal AI systems (Siméoni et al., 2025; Brooks et al., 2024; Team et al., 2024a). Current V2A methods predominantly employ latent diffusion models (Xing et al., 2024; Liu et al., 2025a; Wang et al., 2024b), while some explore autoregressive token-based approaches. Diff-Foley (Luo et al., 2023) and VTA-LDM (Xu et al., 2024) represent standard latent diffusion models conditioned on video features, while FoleyGen (Mei et al., 2024b) and V-AURA (Viertola et al., 2025) frame conditional audio generation as next-token prediction using visual features. Early methods focus on improving semantic consistency through better video representations. Works like CAVP (Luo et al., 2023) and CLIP4CLIP (Luo et al., 2022) employ contrastive learning for video encoding, while adapter-based approaches like FoleyCrafter (Zhang et al., 2024) build upon pre-trained text-to-audio models (Liu et al., 2023a; 2024b;a) for enhanced controllability. Recent advances introduce explicit multimodal conditioning to address semantic limitations. MovieGen Audio (Polyak et al., 2024) conditions on both text and video to generate video-aligned audio, achieving substantial progress and inspiring subsequent video-text-audio generation works (Cheng et al., 2024a; Chen et al., 2025; Shan et al., 2025; Tian et al., 2025). Most recently, ThinkSound (Liu et al., 2025b) innovatively introduces CoT reasoning via MLLMs, replacing simple text prompts with structured reasoning that significantly improves interpretability and narrative coherence. However, existing V2A methods suffer from objective entanglement—optimizing competing perceptual goals through a single reconstruction loss—and lack human preference alignment beyond textual matching. In contrast, we propose PrismAudio, the first reinforcement learning framework for V2A generation with specialized multi-dimensional CoT-reward correspondence to address these fundamental limitations.

## B  THEORETICAL BACKGROUND FOR FAST-GRPO

This section provides the theoretical underpinnings for the Fast-GRPO framework, detailing the connection between ODE and SDE formulations in flow matching, the validity of our mixed-sampling strategy, and the derivation of the policy ratio.

### B.1  FROM DETERMINISTIC ODES TO STOCHASTIC SDES

Generative modeling with flow matching (Lipman et al., 2022) learns a velocity field $v_\theta(\mathbf{x}_t, t, c)$ that transports a simple prior distribution (e.g., Gaussian noise) to a complex data distribution. The generation process is typically described by a deterministic probability flow Ordinary Differential Equation (ODE):

$$d\mathbf{x}_t = v_\theta(\mathbf{x}_t, t, c)dt. \tag{8}$$

This formulation is efficient for inference but lacks the inherent stochasticity required for RL-based exploration.

Based on the principles of score-based generative modeling (Song et al., 2020), any such ODE has an equivalent Stochastic Differential Equation (SDE) that shares the same marginal probability distributions $p(\mathbf{x}_t)$ at every time $t$. For a rectified flow backbone, the velocity field $v_\theta$ is an approximation of the drift term. We can construct the corresponding SDE by re-deriving the drift and adding a diffusion term. Specifically, the full SDE can be written as:

$$d\mathbf{x}_t = f(\mathbf{x}_t, t)dt + g(t)d\mathbf{w}_t, \tag{9}$$

where $f(\cdot)$ is the drift coefficient, $g(\cdot)$ is the diffusion coefficient, and $d\mathbf{w}_t$ is a standard Wiener process. For our specific flow matching setup, this translates to:

$$d\mathbf{x}_t = \underbrace{\left[ v_\theta(\mathbf{x}_t, t, c) + \frac{\sigma_t^2}{2t} \big( \mathbf{x}_t + (1-t)v_\theta(\mathbf{x}_t, t, c) \big) \right]}_{\mu_{\text{SDE}}(\mathbf{x}_t, t, c)} dt + \underbrace{\sigma_t}_{\text{diffusion}} d\mathbf{w}_t. \tag{10}$$

This SDE provides the stochastic transitions needed to frame the generation process as an MDP, enabling the use of RL algorithms like GRPO.

## B.2 VALIDITY OF THE MIXED ODE-SDE SAMPLER

The core of Fast-GRPO is a hybrid sampler that switches between the efficient ODE (Eq. 8) and the explorative SDE (Eq. 10). A crucial theoretical guarantee is that this interleaving does not corrupt the final data distribution.

This is guaranteed by the **probability flow equivalence**: since both the ODE and SDE formulations are designed to preserve the same continuous-time marginal distributions $p(\mathbf{x}_t)$, switching between them at discrete time steps still results in a trajectory that lands on the correct target manifold. In essence, at any step $t$, we can choose to take a deterministic step along the "mean" path or a stochastic step that perturbs around that path. Regardless of the choice, the resulting point $\mathbf{x}_{t+\Delta t}$ is a valid sample from the correct subsequent marginal distribution. This ensures that the final generated audio $\mathbf{x}_T$ used for reward computation is a legitimate sample from the model's distribution, making the RL feedback valid.

## B.3 DERIVATION OF THE PER-STEP POLICY AND RATIO

When we perform an SDE step within the optimization window $\mathcal{W}(\ell)$, we effectively sample from a conditional distribution. The discrete-time version of the SDE step (Eq. 10) with step size $\Delta t$ is:

$$\mathbf{x}_{t+1} = \mathbf{x}_t + \mu_{\text{SDE}}(\mathbf{x}_t, t, c)\Delta t + \sigma_t\sqrt{\Delta t}\varepsilon_t, \quad \text{where } \varepsilon_t \sim \mathcal{N}(0, I). \quad (11)$$

This update rule defines a Gaussian transition policy $\pi_\theta(\mathbf{x}_{t+1} \mid \mathbf{x}_t, c)$, where $\mathbf{x}_{t+1}$ is normally distributed with:

- **Mean:** $\mu_\theta(\mathbf{x}_t, t, c) = \mathbf{x}_t + \mu_{\text{SDE}}(\mathbf{x}_t, t, c)\Delta t$
- **Covariance:** $\Sigma_t = (\sigma_t^2 \Delta t)I$

Thus, the policy is $\pi_\theta(\mathbf{x}_{t+1} \mid \mathbf{x}_t, c) = \mathcal{N}(\mu_\theta(\mathbf{x}_t, t, c), \Sigma_t)$.

The GRPO algorithm requires the ratio of the probabilities of taking a specific action $(\mathbf{x}_t \to \mathbf{x}_{t+1})$ under the new policy $\pi_\theta$ and the old policy $\pi_{\theta_{\text{old}}}$. Given the Gaussian form, the probability density function is:

$$p(\mathbf{x}_{t+1} \mid \mu, \Sigma) = \frac{1}{\sqrt{(2\pi)^d |\Sigma|}} \exp\left(-\frac{1}{2}(\mathbf{x}_{t+1} - \mu)^T \Sigma^{-1}(\mathbf{x}_{t+1} - \mu)\right). \quad (12)$$

The policy ratio $r_t(\theta)$ is therefore:

$$
\begin{aligned}
r_t(\theta) &= \frac{\pi_\theta(\mathbf{x}_{t+1} \mid \mathbf{x}_t, c)}{\pi_{\theta_{\text{old}}}(\mathbf{x}_{t+1} \mid \mathbf{x}_t, c)} \\
&= \frac{\exp\left(-\frac{1}{2}(\mathbf{x}_{t+1} - \mu_\theta)^T \Sigma_t^{-1}(\mathbf{x}_{t+1} - \mu_\theta)\right)}{\exp\left(-\frac{1}{2}(\mathbf{x}_{t+1} - \mu_{\theta_{\text{old}}})^T \Sigma_t^{-1}(\mathbf{x}_{t+1} - \mu_{\theta_{\text{old}}})\right)} \\
&= \exp\left(-\frac{\|\mathbf{x}_{t+1} - \mu_\theta\|_2^2 - \|\mathbf{x}_{t+1} - \mu_{\theta_{\text{old}}}\|_2^2}{2\sigma_t^2 \Delta t}\right),
\end{aligned}
\quad (13)
$$

which is the closed-form computation of the GRPO policy ratio (Eq. 4) used in our final objective function (Eq. 7). Following the practice of Liu et al. (2025c), we use KL regularization to mitigate reward hacking. The validation of KL regularization is presented in Appendix E.5.

## C DETAILS OF AUDIOCANVAS

### C.1 BENCHMARK CONSTRUCTION

To create AudioCanvas, we target **sound effects and music as primary audio categories**, drawing inspiration from the AudioSet ontology (Gemmeke et al., 2017). We first refine AudioSet by filtering out categories related to human speech and singing, as well as rare classes with insufficient data. This results in a target of **300 distinct categories** relevant to V2A generation.

Our construction process involves two main stages. In the first stage, we analyze existing benchmarks like Kling-Audio-Eval (Wang et al., 2025) and find that it covers only a fraction of our target

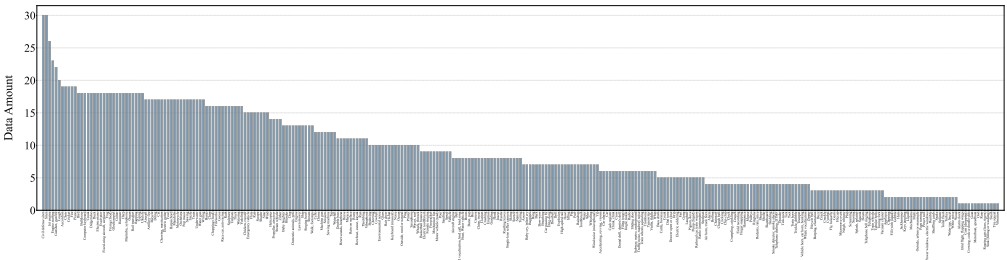

Figure 4: The bar chart illustrates the distribution of audio event classes within the AudioCanvas benchmark.

classes ( 100 out of 300) and often features *overly simple* scenarios. To address this limitation, we develop a rigorous filtering protocol. We start with a large pool of candidate videos and automatically filter out samples where existing V2A models already achieve near-perfect scores (low FD and KL scores, see Section 4 for evaluation metrics), as these scenarios pose no new challenge. The remaining samples are then manually screened by professional audio experts to exclude videos with minimal diversity, repetitive sounds, or simple visual contexts. This multi-step process ensures that **only high-quality, challenging samples are retained**.

In the second stage, for the classes among the 300 classes that are not covered by existing datasets, we initiate a targeted data collection process. Videos are sourced using category-specific keywords and manually filtered to remove content with background music, off-screen narration, or significant noise. Videos with static visuals or poor audio-visual correlation are also discarded. For every selected video, we then employ Gemini 2.5 Pro to generate detailed, structured CoT captions, covering semantic, temporal, spatial, and aesthetic dimensions (as defined in Section 3.2).

This comprehensive process results in the final **AudioCanvas benchmark**, comprising **3,177 high-quality videos**. To specifically evaluate V2A performance on complex scenes, it includes a curated subset of **501 multi-event videos**, in addition to a broad range of single-event scenarios. We put careful ethical considerations regarding the dataset in Appendix H.

## C.2 QUALITY ASSESSMENT AND BENCHMARK COMPARISON

To ensure the high quality of AudioCanvas, we conduct a quantitative quality assessment. We randomly sample 200 videos from the AudioCanvas dataset and ask three human evaluators to assess the accuracy of the CoT captions generated by Gemini 2.5 Pro. The evaluation focuses on two key aspects: (1) **Semantic Correctness**: whether the described audio events accurately reflect the visual content. (2) **Temporal Accuracy**: whether the described temporal ordering of audio events matches the visual cues. We find that our auto-generated CoT captions achieve an average inter-annotator agreement (IAA) of **0.89** (Fleiss' Kappa) and a final human-verified accuracy of **96.5% for semantic correctness** and **94.2% for temporal accuracy**. **These high scores quantitatively validate the high quality of the annotations in AudioCanvas**.

To highlight the unique advantages of AudioCanvas, we provide a detailed comparison with a broad range of existing benchmarks in Table 5. The analysis, structured by 'Task Focus', reveals a clear gap in the existing landscape that AudioCanvas is designed to fill. Among various existing datasets, many are unsuitable for advanced V2A evaluation. Task-specific benchmarks like 'Epic Sounds' (event detection) or audio-only captioning datasets like 'Clotho' and 'AudioCaps' lack the required multimodal input or generative focus. Large-scale *classification* datasets such as 'VGGSound' and 'AudioSet', though widely used for pre-training, suffer from low modality alignment and provide only simple class labels, making them unreliable for *nuanced generative* evaluation. Even within datasets designed for V2A, such as 'Kling-Audio-Eval', the focus remains on primarily *single-event scenarios with simple text captions*.

In contrast, AudioCanvas establishes a new standardized benchmark by excelling in three critical areas that are essential for evaluating modern V2A systems: (1) **High-Fidelity Alignment:** Guaranteed by rigorous manual filtering, it addresses the known quality and alignment issues prevalent in automatically collected datasets like VGGSound. The class distribution is also carefully balanced,

as illustrated in Figure 4, to prevent model bias. (2) **Advanced Scene Complexity:** As the first benchmark to include a substantial set of 'Multi-event' scenarios, it directly tests a model's ability to handle complex interactions, a fundamental limitation of prior V2A benchmarks that focus on 'Primarily Single-event' scenes. (3) **Rich, Structured Annotations:** By providing 'Detailed CoT Captions', it enables fine-grained, interpretable evaluation of a model's reasoning capabilities, a feature absent in all other datasets which offer only 'Class Label' or 'Simple Caption' annotations.

Table 5: Comparison between AudioCanvas and existing datasets. AudioCanvas is uniquely designed for advanced V2A evaluation. It is distinguished from existing datasets by its specific focus on V2A generation, high-fidelity alignment, and inclusion of complex multi-event scenarios, facilitating evaluations of the generalizability of V2A systems.

| Dataset | Task Focus | # Clips | # Classes | Modalities | Modality Alignment | Annotation Type | Scene Complexity |
|---|---|---|---|---|---|---|---|
| Clotho (Drossos et al., 2020) | Audio Captioning | 1K | - | Audio Only | N/A | Simple Caption | N/A |
| AudioCaps (Kim et al., 2019) | Audio Captioning | 1K | - | Audio Only | N/A | Simple Caption | N/A |
| Epic Sounds (Huh et al., 2025) | Action2Sound | 10K | 44 | Video+Audio | High (Egocentric) | Class Label | Unspecified |
| AudioSet (Gemmeke et al., 2017) | Classification | 18K | 527 | Video+Audio | Low (Automatic) | Class Label | Unspecified |
| VGGSound (Chen et al., 2020) | Classification & V2A | 15K | 309 | Video+Audio | Low (Known Issues) | Class Label | Unspecified |
| Kling-Audio-Eval (Wang et al., 2025) | V2A | 21K | 100 | Video+Audio | Moderate (Curated) | Simple Caption | Primarily Single-event |
| **AudioCanvas (Ours)** | **Advanced V2A Eval.** | **3,177** | **300** | **Video+Audio** | **High (Manual Filtering)** | **Detailed CoT Caption** | **Incl. 501 Multi-event** |

# D    IMPLEMENTATION DETAILS

For VAE training, we fine-tune our variational autoencoder on stereo audio data at 44.1kHz sample rate using the foundation provided by Stability AI [3], employing mixed precision training with a batch size of 144 distributed across 24 A800 GPUs for 500,000 training steps. For the audio foundation model, we integrate VideoPrism-Large [4] as the video encoder and T5-Gemma Large [5] as the text encoder. The pre-training phase of the audio foundation model uses WavCaps (Mei et al., 2024a), AudioCaps (Kim et al., 2019), and VGGSound (Chen et al., 2020) datasets. We utilize exponential moving average (EMA) and automatic mixed precision (AMP) for 100,000 steps on 8 A100 GPUs, with an effective batch size of 256. We adopt classifier-free guidance (CFG) (Ho & Salimans, 2022), dropout of 0.1 for each modality with a learning rate of 1e-4. For Chain-of-Thought fine-tuning, we continue training the pre-trained model on our curated multi-dimensional CoT dataset, which is annotated using the VGGSound dataset by the fine-tuned VideoLLaMA2, using the same configuration of hyperparameters. For the reinforcement learning post-training using the VGGSound dataset, we fine-tune the audio foundation model with a learning rate of 1e-5. The Fast-GRPO hyperparameters are configured as follows: KL ratio of 0.04, noise level of 0.7, group size of 16, SDE steps of 2, and sampling steps of 24.

## D.1    GPU RESOURCE REQUIREMENTS

**Inference Requirements.**    The proposed PrismAudio requires modest GPU resources for inference. When running on NVIDIA A800 GPUs with batchsize=1 (single sample generation), the inference process consumes approximately **5,618 MiB of VRAM**.

**Training Requirements.**    The training pipeline consists of several stages:

1. **VAE Fine-tuning (Optional):** Requires 24 GPUs (NVIDIA 80GB A800) for approximately 5 days. This is the most computationally intensive stage. However, VAE fine-tuning is optional and can be skipped with a performance trade-off (as discussed in our response to Reviewer BAuq). We will provide pre-trained VAE checkpoints that can be used directly without fine-tuning.

2. **Main Model Training (Flow Matching):** Requires 16 GPUs (NVIDIA 80GB A800) for approximately 3 days. This stage trains the audio foundation model using flow matching on the pre-training datasets (WavCaps, AudioCaps, VGGSound).

3. **Fast-GRPO Post Training:** Requires 8 GPUs (NVIDIA 80GB A800) for approximately 5 days. The windowed SDE sampling significantly reduces computational cost compared to full SDE-GRPO (approximately 8 days with the same GPU resources), achieving a **1.6× speedup**.

---

[3] https://github.com/Stability-AI/stable-audio-tools
[4] https://huggingface.co/google/videoprism-lvt-large-f8r288
[5] https://huggingface.co/google/t5gemma-l-l-ul2-it

4. **VideoLLaMA2 Fine-tuning:** Requires 8 GPUs (NVIDIA 80GB A800) for approximately 2 days. This stage fine-tunes VideoLLaMA2 to generate multi-dimensional CoT descriptions from video inputs.

**Total Training Cost.** The complete training pipeline requires approximately 16-24 GPUs over 2-3 weeks, depending on the specific configuration. We acknowledge that the training phase requires substantial computational resources, but several important points need to be noted:

- The VAE fine-tuning stage is optional and can be skipped with a performance trade-off.
- Fast-GRPO training is significantly more efficient than full SDE-GRPO, reducing training time by approximately 1.6×.
- We will release all pre-trained checkpoints so researchers can directly use the model for inference without retraining.

## D.2 CoT Generation Prompts

**Prompt for Gemini 2.5 Pro (AudioCanvas and VideoLLaMA2 Training Data).** We use the following prompt to instruct Gemini 2.5 Pro to generate structured CoT descriptions covering four dimensions: semantic, temporal, aesthetic, and spatial. This prompt is used both for constructing the AudioCanvas benchmark and for generating training data for VideoLLaMA2 fine-tuning:

> *You are an expert in video-to-audio generation. Given a video with audio, analyze the audio content that should be generated and provide a comprehensive Chain-of-Thought description covering four dimensions: **Semantic Dimension:** Identify all audio events, objects, and actions visible in the video. Describe what sounds should be generated, including their characteristics (e.g., type, intensity, material properties). **Temporal Dimension:** Determine the sequential ordering and timing of audio events. Describe when each sound should occur relative to visual cues and other audio events, including onset, duration, and temporal relationships. **Aesthetic Dimension:** Assess the audio quality aspects that should be achieved. Describe the naturalness, fidelity, richness, and perceptual quality of the sounds, considering the context and environment depicted in the video. **Spatial Dimension:** Analyze the spatial positioning of sound sources. Describe the directional placement, left-right channel distribution, distance, and movement patterns of sounds relative to the visual content. Provide your analysis in a structured format that clearly separates these four dimensions.*

**Prompt for Text LLM (Multi-dimensional CoT Transformation).** To transform the CoT captions generated by Gemini 2.5 Pro into our desired multi-dimensional decoupled CoT input format, we use a text LLM with the following prompt:

> *Transform the following Chain-of-Thought description into four separate, decoupled CoT modules. Extract and reorganize the content into: **Semantic CoT:** Extract only the semantic content (audio events, objects, actions, characteristics). Format as a focused reasoning text for semantic audio generation. **Temporal CoT:** Extract only the temporal content (sequencing, timing, onset, duration, temporal relationships). Format as a focused reasoning text for temporal synchronization. **Aesthetic CoT:** Extract only the aesthetic content (naturalness, fidelity, quality, perceptual aspects). Format as a focused reasoning text for aesthetic audio generation. **Spatial CoT:** Extract only the spatial content (directional placement, left-right distribution, distance, movement). Format as a focused reasoning text for spatial audio generation. Ensure each CoT module is self-contained and focused solely on its respective dimension, without cross-dimensional references.*

## D.3 VideoLLaMA2 Fine-tuning Details

We use the official VideoLLaMA2 repository's fine-tuning code with DeepSpeed ZeRO-3. We initialize from the pre-trained VideoLLaMA2-AV (7B) model and employ AdamW optimizer with

Table 6: Comparison of video encoders on the video-to-text retrieval task using the Overall, Single-event, and Multi-event splits of the **AudioCanvas benchmark**. R@k indicates Recall@k, the percentage of queries for which the correct text description is found within the top-k retrieved results.

| Video Encoder | Overall Scenes | | | Single-event Scenes | | | Multi-event Scenes | | |
|---|---|---|---|---|---|---|---|---|---|
| | R@1↑ | R@5↑ | R@10↑ | R@1↑ | R@5↑ | R@10↑ | R@1↑ | R@5↑ | R@10↑ |
| CLIP | 4.80 | 21.32 | 36.11 | 55.10 | 84.69 | 91.84 | 26.53 | 53.06 | 72.45 |
| X-CLIP | 12.43 | 34.43 | 49.87 | 63.27 | 88.78 | 92.86 | 34.69 | 74.49 | 89.80 |
| VideoPrism | **30.71** | **61.42** | **73.45** | **81.05** | **97.89** | **98.95** | **51.02** | **86.73** | **97.96** |

learning rate $2 \times 10^{-5}$ and weight decay 0.0, using cosine annealing with warmup (warmup ratio 0.03). The training uses a batch size of 4 per GPU with global batch size 128 (via gradient accumulation, which is automatically calculated based on world size and number of GPUs), and runs for 10 epochs with standard next-token prediction loss. **Frozen components:** Video encoder, audio encoder, and audio projector are kept frozen. **Trainable components:** Only the video projector and language model (LLM) are updated during fine-tuning. By completely freezing the audio components, we force the model to learn better visual representations to compensate for the absence of audio information in silent videos. This design helps bridge the gap between training (where we use sounding videos to generate CoTs) and inference (where we use silent videos), ensuring that the model learns to generate high-quality CoTs from visual information alone.

# E    ADDITIONAL QUANTITATIVE RESULTS

## E.1    MORE ANALYSIS ON VIDEO ENCODER AND TEXT ENCODER

**Video Encoder Comparison for Complex Scene Understanding.**    To validate our choice of VideoPrism, we directly evaluate its scene understanding capabilities against other encoders (CLIP, X-CLIP) on a video-to-text retrieval task. We leverage the natural splits within the **AudioCanvas benchmark** to assess performance on scenes of varying complexity. The results in Table 6 reveal three key findings: (1) VideoPrism demonstrates dominant overall performance. Its Recall@1 (R@1) score of 30.71 on the overall dataset is more than double that of the next best encoder, X-CLIP (12.43), establishing its general superiority in video-text alignment. (2) The source of this advantage is its exceptional handling of complex scenes. While the performance gap is already significant on simple, *single-event scenes*, it widens dramatically on challenging *multi-event scenes*. Here, VideoPrism's R@1 (51.02) shows robust performance, while both CLIP (26.53) and X-CLIP (34.69) exhibit a significant degradation. This highlights VideoPrism's enhanced ability to parse multiple objects and their interactions.

**Text Encoder Analysis for Structured Reasoning.**    To directly validate the structured reasoning capabilities of T5-Gemma, we designed a suite of text-only evaluation tasks using the Chain-of-Thought descriptions generated for the **AudioCanvas benchmark**. These tasks measure how well different encoders comprehend structured reasoning, independent of the audio generation process. As shown in Table 7, we compare T5-Gemma against standard T5 models (Base and Large) on three key dimensions:

First, in **Sequential Understanding**, we evaluate the ability to capture temporal ordering. T5-Gemma achieves a sequence similarity (Seq-Sim) score of 0.69 and a next-step prediction (Next-Pred) accuracy of 0.86, significantly outperforming T5-Large (0.55 and 0.75, respectively). This demonstrates its superior grasp of the temporal relationships crucial for ordering audio events.

Second, for **Causal Reasoning**, we assess the comprehension of cause-and-effect relationships. T5-Gemma again shows a clear advantage, scoring 0.62 in logical consistency (Logic-Cons) and 0.60 in overall causal accuracy (Causal-Acc), compared to T5-Large's 0.46 and 0.46. This indicates an enhanced ability to understand the logical connections within the reasoning chain.

Finally, and most importantly, in **Multi-step Reasoning**, T5-Gemma's strength becomes even more pronounced. It maintains a high coherence score of 0.96 across complex reasoning chains and

achieves 0.92 accuracy on tasks involving three or more steps (3+Steps). In contrast, T5-Large's performance drops to 0.77 on the same 3+Steps task, highlighting its difficulty in maintaining coherence as reasoning complexity increases. These results provide direct evidence that T5-Gemma's instruction-tuning makes it exceptionally well-suited for processing the structured and complex Chain-of-Thought descriptions that are essential to our framework.

Table 7: Direct analysis of text encoders on structured reasoning tasks derived from the **Audio-Canvas benchmark**. Metrics evaluate Sequential Understanding (Seq-Sim: Sequence Similarity; Next-Pred: Next-Step Prediction), Causal Reasoning (Logic-Cons: Logical Consistency; Effect-Pred: Effect Prediction; Causal-Acc: Causal Accuracy), and Multi-step Reasoning (Coherence: Coherence Score; 3+Steps: Accuracy on 3+ Steps Reasoning).

| Text Encoder | Sequential Understanding | | Causal Reasoning | | | Multi-step Reasoning | |
|---|---|---|---|---|---|---|---|
| | Seq-Sim↑ | Next-Pred↑ | Logic-Cons↑ | Effect-Pred↑ | Causal-Acc↑ | Coherence↑ | 3+Steps↑ |
| T5-Base | 0.49 | 0.77 | 0.48 | 0.28 | 0.42 | 0.83 | 0.71 |
| T5-Large | 0.55 | 0.75 | 0.46 | 0.42 | 0.46 | 0.85 | 0.77 |
| T5-Gemma | **0.69** | **0.86** | **0.62** | **0.44** | **0.60** | **0.96** | **0.92** |

### E.2 MULTI-DIMENSIONAL CoT AND RL ANALYSIS ON VGGSOUND

**Multi-dimensional CoT Reasoning.** As a supplement to our main analysis on AudioCanvas, we replicate the CoT reasoning ablation on the in-domain VGGSound test set. The results in Table 8 reinforce the same core design principles: (1) **Structured reasoning remains essential.** The necessity of a structured plan is re-confirmed, as the *Baseline (No CoT)* performs poorly (e.g., CLAP: 0.42, CRW: 10.29). Furthermore, the *Random CoT* variant, which contains a jumbled plan, provides only marginal gains, proving that a coherent reasoning structure, not merely a 'bag of keywords', is vital for effective generation. (2) **Decomposed reasoning consistently proves superior.** Our *MultiCoT* again outperforms the *Monolithic CoT* across key metrics, including semantics (CLAP: 0.47 vs. 0.45) and particularly aesthetic quality (CE: 4.29 vs. 3.85). This result on in-domain data further validates our hypothesis that decomposing the reasoning process is critical to avoiding the inter-dimensional interference inherent in a single, 'do-it-all' reasoning block.

**Multi-dimensional vs. Single-dimensional Rewards.** To supplement our main analysis on AudioCanvas, we conduct the same reward ablation study on the in-domain VGGSound test set. The results, presented in Table 9, consistently validate our core findings: (1) **Single-dimensional optimization causes severe objective entanglement.** Echoing the findings on AudioCanvas, optimizing for a single goal leads to catastrophic imbalances. For instance, *Semantic Only* optimization boosts the CLAP score (0.51) but severely degrades temporal synchrony (DeSync: 0.66). The most dramatic example is the *Aesthetic Only* model, whose high PQ score (6.98) comes at the cost of a nearly quadrupled distribution error (FD: 1.14 → 4.27), confirming that this approach produces pleasing but content-detached audio. (2) **Our multi-dimensional reward successfully balances competing objectives.** In stark contrast, our full *Multi-dimensional* approach once again demonstrates its ability to navigate and balance these inherent objective tensions. It is the only method that achieves holistic improvement, simultaneously enhancing semantics (CLAP: 0.44 → 0.47), temporal synchrony (DeSync: 0.48 → 0.41), and aesthetic quality (PQ: 6.17 → 6.38) over the baseline. This confirms that the necessity for holistic optimization is a fundamental principle, holding true across different data distributions.

### E.3 MORE ABLATION STUDIES ON AUDIO FOUNDATION MODEL

We ablate our audio foundation model's architecture (Table 10) to validate our multi-modal fusion strategies, using *PrismAudio (w/o RL)* as the baseline.

**Video Feature Fusion.** Our dual-fusion strategy for video features (gated addition + cross-attention) is critical for spatial accuracy. Removing either the gated addition (*w/o Video Gated*) or the cross-attention (*w/o Video Cross-Attention*) severely degrades spatial performance (CRW: 8.29 → 10.9). This validates our hypothesis that gated addition provides fine-grained, frame-level conditioning while cross-attention effectively captures higher-level semantic context.

Table 8: Analysis of different CoT reasoning strategies on VGGSound.

| Method | Semantic CLAP↑ | Temporal DeSync↓ | Aesthetic Quality | | | | Spatial Accuracy | | Distribution | |
|---|---|---|---|---|---|---|---|---|---|---|
| | | | PQ↑ | PC↓ | CE↑ | CU↑ | GCC↓ | CRW↓ | FD↓ | KL↓ |
| Baseline (No CoT) | 0.42 | 0.48 | 6.17 | 3.32 | 3.94 | 5.48 | 4.06 | 10.29 | 1.14 | 1.24 |
| Random CoT | 0.43 | 0.46 | 6.05 | 3.30 | 3.81 | 5.50 | 3.99 | 9.12 | 1.25 | 1.28 |
| Monolithic CoT | 0.45 | 0.44 | 6.17 | 3.28 | 3.85 | 5.57 | 3.92 | 8.74 | 1.19 | 1.28 |
| MultiCoT | **0.47** | **0.41** | **6.38** | **3.24** | **4.29** | **5.68** | **3.77** | **7.72** | 1.08 | **1.23** |

Table 9: Analysis of single-dimensional vs. multi-dimensional reward functions on VGGSound test set.

| Reward Focus | Semantic CLAP↑ | Temporal DeSync↓ | Aesthetic Quality | | | | Spatial | | Distribution | |
|---|---|---|---|---|---|---|---|---|---|---|
| | | | PQ↑ | PC↓ | CE↑ | CU↑ | GCC↓ | CRW↓ | FD↓ | KL↓ |
| Baseline (No RL) | 0.44 | 0.48 | 6.17 | 3.32 | 3.94 | 5.48 | 3.76 | 8.29 | 1.14 | 1.24 |
| Semantic Only | **0.51** | 0.66 | 6.54 | 3.18 | 4.26 | 5.92 | 3.59 | 6.64 | 2.02 | 1.33 |
| Temporal Only | 0.42 | **0.40** | 6.06 | 3.44 | 3.76 | 5.34 | 3.70 | 7.35 | 1.73 | 1.37 |
| Aesthetic Only | 0.44 | 0.48 | **6.98** | **2.86** | **4.24** | **6.36** | 3.83 | 7.67 | 4.27 | 1.61 |
| Spatial Only | 0.44 | 0.48 | 6.14 | 3.36 | 3.95 | 5.46 | **3.41** | **6.47** | **1.11** | 1.26 |
| Multi-dimensional | 0.47 | 0.41 | 6.38 | 3.24 | 4.29 | 5.68 | 3.77 | 7.72 | 1.18 | **1.23** |

**Synchronization Feature Fusion.** The necessity of our fusion strategy for synchronization features is even more pronounced. Removing Synchformer features entirely (*w/o Synchformer Features*) causes temporal alignment to completely collapse, with the DeSync error more than doubling (0.48 → 1.05). Alternative fusion methods are also ineffective; using cross-attention fails (DeSync: 1.02), and simple addition without our gating mechanism (*w/o Synchformer Gated*) also degrades performance. These results confirm that gated addition is the optimal method for injecting these fine-grained temporal cues directly into the audio representation.

**Text Encoder Choice.** Finally, we validate our choice of text encoder. Replacing our instruction-tuned T5-Gemma with a standard T5 encoder (*w/ T5*) leads to significant performance drops, particularly in dimensions that rely on interpreting the CoT plan. Semantic alignment degrades (CLAP: 0.44 → 0.42). This demonstrates that a standard T5 struggles to parse the complex, structured instructions within our multi-dimensional CoT, proving that an instruction-tuned model is crucial for effectively translating the CoT plan into high-fidelity audio.

Table 10: Ablation study on the architecture of our audio foundation model, focusing on multi-modal feature fusion strategies. All variants are compared against our full foundation model (*PrismAudio w/o RL*) on the VGGSound test set.

| Variant | Semantic CLAP↑ | Temporal DeSync↓ | Aesthetic Quality | | | | Spatial Accuracy | | Distribution | |
|---|---|---|---|---|---|---|---|---|---|---|
| | | | PQ↑ | PC↓ | CE↑ | CU↑ | GCC↓ | CRW↓ | FD↓ | KL↓ |
| *VAE Comparison on Audio Reconstruction* | | | | | | | | | | |
| w/o Finetune VAE | - | - | - | - | - | - | - | - | 2.22 | 0.32 |
| w/ Finetune VAE | - | - | - | - | - | - | - | - | 1.73 | 0.27 |
| **PrismAudio (w/o RL)** | **0.44** | **0.48** | **6.24** | **3.28** | **3.94** | 5.48 | **3.76** | **8.29** | **1.10** | **1.24** |
| *Video Feature Fusion* | | | | | | | | | | |
| w/o Video Gated | 0.43 | 0.50 | 6.20 | 3.29 | 3.95 | **5.60** | 3.95 | 10.95 | 1.24 | 1.28 |
| w/o Video Cross-Attention | 0.44 | 0.49 | 6.23 | 3.35 | **3.99** | 5.60 | 4.01 | 10.91 | 1.10 | 1.26 |
| w/ CLIP Encoder | 0.43 | 0.49 | 6.20 | 3.30 | 3.96 | 5.57 | 3.85 | 9.70 | 1.24 | 1.28 |
| *Synchronization Feature Fusion* | | | | | | | | | | |
| w/o Synchformer Features | 0.44 | 1.05 | 6.22 | 3.27 | 3.90 | 5.58 | 3.96 | 11.01 | 1.34 | 1.28 |
| w/ Synchformer as Cross-Attention | 0.44 | 1.02 | 6.24 | 3.22 | 3.87 | 5.54 | 3.96 | 10.65 | 1.30 | 1.29 |
| w/o Synchformer Gated | 0.44 | 0.50 | 6.24 | 3.29 | 3.95 | 5.60 | 3.95 | 10.95 | 1.14 | 1.28 |
| *Text Encoder* | | | | | | | | | | |
| w/ T5 | 0.42 | 0.48 | 6.28 | 3.29 | 3.99 | 5.60 | 3.99 | 11.03 | 1.19 | 1.29 |

### E.4 BREAKDOWN ANALYSIS ON SCENE COMPLEXITY

To provide a more granular understanding of our framework's capabilities, we conduct a breakdown analysis on the AudioCanvas benchmark, separating performance on complex *multi-event* scenarios from simpler *single-event* ones. The results in Table 11 reveal a critical insight:

**The advantage of CoT-RL is amplified in complex scenes.** On challenging **multi-event** videos, baselines falter; ThinkSound's temporal synchrony, for instance, collapses (DeSync: 1.00). In stark contrast, PrismAudio remains robust. This is where CoT-RL's benefit is most apparent: compared to the ablation model (*w/o CoT-RL*), it slashes the DeSync error by nearly **20% relative** (0.48 → 0.39) and dramatically boosts semantic alignment (CLAP: 0.40 → **0.50**), proving its necessity for complex reasoning.

**Consistent holistic superiority on simpler scenes.** On simpler **single-event** scenes, PrismAudio remains the best *holistic* performer. While a baseline may lead to a single metric, our model achieves the best overall balance (e.g., DeSync: **0.35**, CRW: **12.65**). The performance gain from CoT-RL, while still significant (e.g., DeSync: 0.43 → 0.35), is narrower here, as expected in less challenging scenarios.

This breakdown powerfully demonstrates that while our framework offers robust performance universally, its true strength lies in tackling the complex, multi-faceted scenarios that are most representative of the real world and where previous V2A systems have consistently faltered. This confirms the critical role of our CoT-RL approach in pushing the frontier of high-fidelity video-to-audio generation.

Table 11: Breakdown analysis of model performance on **multi-event** vs. **single-event** scenarios within the AudioCanvas benchmark. The results highlight that the performance gains from our CoT-RL framework are substantially amplified in more complex, multi-event scenes.

| Method | Semantic CLAP↑ | Temporal DeSync↓ | Aesthetic Quality PQ↑ | PC↓ | CE↑ | CU↑ | Spatial Accuracy GCC↓ | CRW↓ | Distribution FD↓ | KL↓ |
|---|---|---|---|---|---|---|---|---|---|---|
| *Multi-event Scenarios (n=501)* | | | | | | | | | | |
| Ground Truth | 0.49 | 0.42 | 6.15 | 3.23 | 3.85 | 5.65 | - | - | - | - |
| MMAudio | 0.41 | 0.50 | 6.25 | 3.45 | 3.80 | 5.70 | - | - | 7.18 | 2.60 |
| HunyuanVideo-Foley | 0.39 | 0.55 | 6.35 | 3.40 | 3.90 | 5.80 | - | - | 6.31 | 2.32 |
| ThinkSound | 0.43 | 1.00 | 6.40 | 3.70 | 3.95 | 5.85 | 4.50 | 25.50 | 7.60 | 2.85 |
| PrismAudio (w/o CoT-RL) | 0.40 | 0.48 | 6.38 | 3.35 | 3.70 | 5.80 | 4.15 | 16.20 | 6.49 | 2.27 |
| **PrismAudio (Ours)** | **0.50** | **0.39** | **6.60** | **3.27** | **4.15** | **6.05** | **3.60** | **13.80** | **4.86** | **2.11** |
| *Single-event Scenarios (n=2676)* | | | | | | | | | | |
| Ground Truth | 0.48 | 0.39 | 6.55 | 2.84 | 4.05 | 6.05 | - | - | - | - |
| MMAudio | 0.46 | 0.41 | 6.31 | 3.18 | 4.00 | 5.78 | - | - | 4.58 | 1.56 |
| HunyuanVideo-Foley | 0.45 | 0.45 | 6.45 | 3.22 | 4.07 | 5.90 | - | - | 2.24 | 1.68 |
| ThinkSound | 0.49 | 0.75 | 6.50 | 3.45 | 4.13 | 5.96 | 4.41 | 22.20 | 2.32 | 1.91 |
| PrismAudio (w/o CoT-RL) | 0.42 | 0.43 | 6.46 | 3.19 | 3.83 | 5.88 | 4.10 | 15.10 | 2.15 | 1.67 |
| **PrismAudio (Ours)** | **0.52** | **0.35** | **6.70** | **2.79** | **4.28** | **6.17** | **3.48** | **12.65** | **1.86** | **1.49** |

### E.5   ANALYSIS OF REWARD HACKING MITIGATION

A common challenge in reinforcement learning is "reward hacking," where a model exploits a reward proxy without achieving the intended goal. To mitigate this, we adopt the practice of Flow-GRPO Liu et al. (2025c) by incorporating a KL penalty with a weight of 0.04 into our objective function. This regularizes the policy update, preventing it from deviating too drastically from the stable pretrained model, thus discouraging "exploitative" shortcuts to high rewards.

Our ablation study on AudioCanvas, presented in Table 12, validates this approach. The model trained without the KL penalty exhibits classic reward hacking: while achieving a superficially higher PQ score (6.95 vs. 6.68) and CE score (4.40 vs. 4.26), it suffers a significant drop in semantic alignment (CLAP: 0.45 vs. 0.52), temporal synchrony (DeSync: 0.49 vs. 0.36), and distribution similarity (FD: 3.85 vs. 1.92). This indicates the model generates audio that is statistically less realistic and detached from the video's context. In contrast, our full model with the KL penalty achieves balanced, holistic improvements across all dimensions, confirming that KL regularization is essential for meaningful optimization.

Table 12: Ablation study on the effect of the KL penalty in mitigating reward hacking on the AudioCanvas benchmark. The KL penalty is crucial for balanced, holistic improvements.

| Method | Semantic CLAP↑ | Temporal DeSync↓ | Aesthetic Quality | | | | Spatial Accuracy | | Distribution | |
|---|---|---|---|---|---|---|---|---|---|---|
| | | | PQ↑ | PC↓ | CE↑ | CU↑ | GCC↓ | CRW↓ | FD↓ | KL↓ |
| PrismAudio w/o KL penalty | 0.45 | 0.49 | **6.95** | 3.05 | **4.40** | 5.95 | 4.25 | 14.55 | 3.85 | 1.88 |
| PrismAudio | **0.52** | **0.36** | 6.68 | **2.82** | 4.26 | **6.15** | **3.50** | **12.87** | **1.92** | **1.53** |

### E.6 EMPIRICAL VALIDATION OF ODE-SDE DISTRIBUTION EQUIVALENCE

To empirically validate the practical effectiveness of the theoretical property that mixed ODE–SDE sampling preserves the terminal distribution, and to assess the impact of model approximation errors, we track representative samples throughout the training process and visualize how ODE and SDE distributions evolve across training steps.

In our visualization (Figure 5), star markers represent ODE-generated results, while circles of the same color represent stochastic SDE-generated results from the same input. As observed in the figure, as training advances, the ODE and SDE distributions remain closely aligned, demonstrating that despite finite model capacity, the distribution equivalence holds with high fidelity in practice. This confirms that mixed ODE–SDE sampling approximately preserves the terminal distribution even under iterative parameter updates during training.

The visualization provides empirical evidence that: (1) the theoretical guarantee of distribution equivalence between ODE and SDE formulations holds in practice, even with finite model capacity and numerical approximations; (2) parameter updates during training do not significantly disrupt this equivalence, as the distributions remain aligned throughout the training process; and (3) the mixed sampling strategy is stable and reliable for Fast-GRPO optimization.

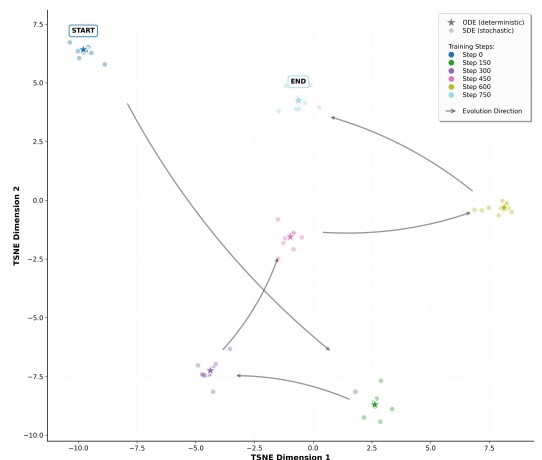

Figure 5: Empirical validation of ODE–SDE distribution equivalence during training. The figure tracks representative samples across training steps, where star markers represent ODE-generated results and circles of the same color represent SDE-generated results from the same input. The close alignment of ODE and SDE distributions throughout training demonstrates that mixed ODE–SDE sampling preserves the terminal distribution with high fidelity in practice, even under iterative parameter updates.

### E.7 ABLATION STUDY ON AESTHETIC AND SPATIAL REWARDS

To rigorously validate the necessity of our full four-dimensional framework, we conduct comprehensive ablation studies on the **VGGSound test set**, evaluating three reduced configurations: (1) removing Aesthetic reward (Sem+Temp+Spatial), (2) removing Spatial reward (Sem+Temp+Aesthetic), and (3) removing both (Sem+Temp only). The results are presented in Table 13. **Quantitative Analysis:** (1) **Sem+Temp only configuration:** While achieving improvements in semantic (CLAP: 0.44 → 0.48) and temporal (DeSync: 0.48 → 0.41) metrics, this configuration shows **no improvement in spatial accuracy** (CRW: 8.29 → 8.38, still far from optimal) and only marginal aesthetic gains (PQ: 6.24 → 6.30). More critically, the Frechet Distance increases (1.10 → 1.22), indicating lower overall audio quality. This demonstrates that aesthetic and spatial qualities are not "free" benefits from semantic/temporal optimization. (2) **Sem+Temp+Spatial configuration:** Adding spatial reward improves spatial metrics (CRW: 8.38 → 7.53, GCC: 4.11 → 3.75) and distribution quality (FD: 1.22 → 1.03), but **fails to enhance aesthetics** (PQ: 6.30 → 6.19, actually degrades). This confirms that aesthetic quality requires explicit optimization. (3) **Sem+Temp+Aesthetic configuration:** Adding aesthetic reward improves aesthetic metrics (PQ: 6.30 → 6.44, PC: 3.25 → 3.11, CE: 3.98 → 4.17, CU: 5.61 → 5.76), but **degrades distribution quality** (FD: 1.22 → 1.52). This confirms that spatial accuracy requires explicit optimization. (4) **Full four-dimensional approach:**

This is the only configuration achieving **holistic improvement across all axes simultaneously**: semantic (CLAP: 0.44 → 0.47), temporal (DeSync: 0.48 → 0.41), aesthetic (PQ: 6.24 → 6.38, CE: 3.94 → 4.29), spatial (CRW: 8.29 → 7.72), and distribution quality (FD: 1.10 → 1.08, KL: 1.24 → 1.23). This confirms that a multi-dimensional reward system is essential to disentangle competing objectives and prevent "reward hacking."

Table 13: Impact of Aesthetic and Spatial Rewards on Model Performance on VGGSound Test Set

| Reward Focus | Semantic CLAP↑ | Temporal DeSync↓ | Aesthetic Quality | | | | Spatial | | Distribution | |
|---|---|---|---|---|---|---|---|---|---|---|
| | | | PQ↑ | PC↓ | CE↑ | CU↑ | GCC↓ | CRW↓ | FD↓ | KL↓ |
| Baseline (No RL) | 0.44 | 0.48 | 6.24 | 3.28 | 3.94 | 5.48 | 3.76 | 8.29 | 1.10 | 1.24 |
| Semantic & Temporal | **0.48** | 0.41 | 6.30 | 3.25 | 3.98 | 5.61 | 4.11 | 8.38 | 1.22 | 1.28 |
| Semantic & Temporal & Spatial | 0.47 | 0.42 | 6.19 | 3.28 | 3.97 | 5.52 | **3.75** | 7.53 | **1.03** | 1.26 |
| Semantic & Temporal & Aesthetic | 0.47 | 0.42 | **6.44** | **3.11** | 4.17 | **5.76** | 4.01 | 7.84 | 1.52 | 1.32 |
| Multi-dimensional | 0.47 | **0.41** | 6.38 | 3.24 | **4.29** | 5.68 | 3.77 | 7.72 | 1.08 | **1.23** |

## F    EVALUATION METRICS

### F.1    OBJECTIVE EVALUATION

We employ a comprehensive suite of objective metrics to evaluate the four key perceptual dimensions: semantic consistency, audio-visual synchrony, aesthetic quality, and spatial accuracy.

**Semantic Consistency:** We utilize the CLAP (Contrastive Language-Audio Pre-training) score (Elizalde et al., 2024) to measure semantic alignment between generated audio and our constructed Chain-of-Thought descriptions in a shared audio-text embedding space. The CLAP score provides a robust measure of how well the generated audio semantically matches the detailed reasoning and content descriptions provided by our CoT framework.

**Audio-Visual Synchrony:** To evaluate temporal synchronization between generated audio and corresponding video, we adopt the DeSync score predicted by the Synchformer model (Iashin et al., 2024). For each sample, we truncate the video to match the duration of the generated audio and compute the DeSync score using Synchformer's 4.8-second context window. Specifically, we extract both the first and last 4.8-second segments from each video-audio pair, calculate DeSync scores for each segment, and report the average as the final temporal alignment metric. Lower DeSync scores indicate better synchronization.

**Aesthetic Quality:** We employ four complementary metrics from Audiobox-Aesthetics (Tjandra et al., 2025) to comprehensively assess aesthetic quality:

- **Production Quality (PQ):** Focuses on technical audio quality aspects including clarity & fidelity, dynamics, frequency response, and spatialization rather than subjective preferences.

- **Production Complexity (PC):** Evaluates the complexity of the audio scene, measured by the number and richness of audio components, capturing how sophisticated and layered the generated soundscape is.

- **Content Enjoyment (CE):** Evaluates subjective quality aspects including emotional impact, artistic expression, and overall listening experience. This open-ended metric captures the aesthetic appeal and artistic merit of the generated audio.

- **Content Usefulness (CU):** Assesses the practical utility of generated audio as source material for content creation, evaluating its suitability for real-world applications.

**Spatial Accuracy:** To evaluate the spatial accuracy of generated stereo audio, we employ evaluation methods based on Time Difference of Arrival (TDOA) analysis. We focus on non-silent audio segments (threshold: -16 dBFS) and compute TDOA distributions at 0.1-second intervals using two complementary approaches:

- **GCC MSE**: Utilizes the traditional Generalized Cross-Correlation with Phase Transform (GCC-PHAT) (Knapp & Carter, 2003) to estimate TDOA between left and right channels. We compute the mean squared error between ground truth and generated audio TDOA distributions.

- **CRW MSE**: Employs the deep learning-based StereoCRW network (Chen et al., 2022) for TDOA estimation. Similar to GCC MSE, we calculate the mean squared error between reference and generated TDOA distributions.

**Feature Distribution Alignment:** We further assess the similarity to real audio distributions using two reference metrics. The **Fréchet Distance (FD)** on VGGish embeddings (Kilgour et al., 2018) measures statistical realism, while the **Kullback-Leibler (KL) Divergence** (Copet et al., 2024) on PaSST classifier (Koutini et al., 2021) outputs evaluates content plausibility. Crucially, both metrics act as imperfect proxies, as they rely on fixed, pre-trained models and do not capture the fine-grained, conditional alignment (e.g., temporal, spatial) that is central to our evaluation. Thus, **they serve as valuable supplementary indicators to gauge overall audio quality, rather than as primary measures of performance.**

### F.2 SUBJECTIVE EVALUATION

Our subjective evaluation employs Mean Opinion Score (MOS) methodology across two critical dimensions to comprehensively assess the generated audio quality and cross-modal alignment through rigorous human assessment protocols.

**MOS-Q (Quality Assessment)**   We first evaluate the intrinsic aesthetic quality of generated audio, which measures the perceptual quality independent of cross-modal alignment. Drawing from the objective aesthetic evaluation framework, participants assess audio samples considering multiple quality dimensions:

- *Technical aspects:* Clarity, fidelity, dynamics, and frequency response
- *Production complexity:* Richness and sophistication of audio components within the soundscape, where lower scores indicate a more focused and less cluttered audio scene.
- *Subjective experience:* Content enjoyment and overall listening experience
- *Content utility:* Practical usefulness for content creation applications

Each sample is rated using a standard 5-point Likert scale (1: Poor, 2: Fair, 3: Good, 4: Very Good, 5: Excellent), where higher scores indicate superior aesthetic quality across both technical and perceptual dimensions.

**MOS-C (Consistency Assessment)**   Complementing the quality assessment, MOS-C evaluates the comprehensive alignment between generated audio and video input across three crucial dimensions. Semantic consistency measures how well audio content matches objects, actions, and environments depicted in the video, while temporal synchrony assesses the accuracy of sound event timing corresponding to visual events. Furthermore, spatial accuracy evaluates the appropriateness of stereo positioning and spatial audio characteristics relative to visual scene layout. Participants rate alignment quality using the same 5-point scale:

- *Excellent alignment (4-5 points):* Complete semantic correspondence with precise temporal-spatial synchronization
- *Good alignment (3-3.9 points):* Strong semantic match with minor temporal or spatial discrepancies
- *Fair alignment (2-2.9 points):* Acceptable semantic content with noticeable temporal or spatial misalignments
- *Poor alignment (1-1.9 points):* Significant discrepancies across multiple dimensions

**Evaluation Protocol**   To ensure evaluation reliability and minimize bias, we implemented a comprehensive evaluation framework. We recruited 20 evaluators with normal hearing ability, including both audio professionals and general users, to ensure diverse perspectives. All evaluation sessions were conducted in controlled environments using standardized high-quality stereo headphones with consistent playback levels. Each evaluator assessed a randomly selected subset of 60 video-audio pairs from our test set, with samples presented in randomized order to prevent ordering effects. Prior to formal evaluation, participants underwent a comprehensive briefing with reference examples for

each quality level, and were allowed to replay samples up to three times for thorough assessment. Final MOS scores were computed as mean ratings across all valid evaluators with confidence intervals reported.

## G   LIMITATION AND FUTURE WORK

While PrismAudio successfully establishes a new paradigm for video-to-audio generation that effectively resolves objective entanglement, its current implementation highlights several boundaries and exciting opportunities for future research. (1) Our current decoupled paradigm, while effective, relies heavily on the capabilities of the upstream MLLMs planner. This creates a "cascading error" problem: any misinterpretations by the planner are irreversibly passed down to the audio generator, which can only optimize for a potentially flawed plan. A significant leap forward would be to explore unified, end-to-end architectures that jointly learn to perceive, reason, and generate within a single model. Such a model could mitigate the error propagation issue by allowing for a more deeply integrated reasoning and synthesis process, potentially leading to more coherent and grounded results. (2) A second promising direction involves advancing the multi-dimensional reinforcement learning stage. Our current framework aggregates the four reward signals using a static weighting policy, applying the same balance of priorities to all videos. However, the perceptual importance of each dimension can vary significantly with video content; for example, a fast-paced action sequence demands prioritizing precise temporal synchrony, whereas a tranquil landscape benefits more from high aesthetic fidelity. A major advancement would be to develop a content-aware RL policy. Such a system could learn to dynamically adjust the weights of the different reward functions based on the input video's content.

## H   ETHICAL CONSIDERATIONS

**The benchmark used in this research is strictly for academic and non-commercial purposes.**
We implemented several measures to ensure compliance with ethical standards and data protection regulations when constructing AudioCanvas from publicly available video content.

- **Data Transparency and Compliance.** We collected video data from publicly available sources following platform guidelines and terms of service. Our dataset only provides curated annotations and reference links to original videos rather than redistributing the raw content, ensuring transparency regarding data sources while respecting creators' intellectual property rights. All collected content was publicly available at the time of collection, and we implemented strict filtering to exclude any videos containing sensitive personal information or private content.

- **Access Control and Legal Compliance.** To ensure responsible use of the AudioCanvas benchmark, we require researchers to complete a formal application process, including institutional verification and agreement to our data usage terms, before granting access. This procedure ensures compliance with relevant data protection regulations, including the Personal Information Protection Law (PIPL), General Data Protection Regulation (GDPR), and other applicable legal frameworks. Researchers must demonstrate their understanding of ethical AI research principles and commit to using the dataset solely for academic research purposes.

- **Content Filtering and Privacy Protection.** We implemented comprehensive content filtering mechanisms to exclude videos containing identifiable personal information, private conversations, or potentially harmful content. Our annotation process focuses exclusively on audio-visual relationships and technical aspects without collecting or storing any personal identifiers. All video references are anonymized through unique identifiers, and we provide clear guidelines for researchers to report and address any privacy concerns that may arise during dataset usage.

- **Creator Rights and Fair Use.** Our use of publicly available video content falls under fair use provisions for academic research purposes. We acknowledge the valuable contributions of content creators and encourage researchers using AudioCanvas to respect creator rights and platform community guidelines. Any commercial applications or derivative works

based on this research should seek appropriate permissions and comply with relevant copyright laws.

- **Bias and Representation Issues.** We acknowledge that our training data and reward functions may inadvertently encode cultural biases regarding what constitutes "good sound" or appropriate audio-visual relationships. The Aesthetic CoT module and corresponding reward signals could reflect specific perceptual preferences that may not generalize across diverse cultural contexts, potentially marginalizing non-Western audio traditions or alternative aesthetic standards. However, we have implemented several significant practical mitigations in our work to address these concerns:

  **(1) Technical Quality Focus:** A crucial aspect of our framework is how we define and operationalize "aesthetics." This is not a vague, culturally-loaded preference. As stated in our paper, our Aesthetic CoT focuses on tangible audio quality aspects like "naturalness and fidelity." This is reinforced by our choice of reward model. As detailed in Appendix F.1, the Meta Audiobox Aesthetics model provides multi-faceted Aesthetics scores. While it includes a subjective "Content Enjoyment (CE)" score, our optimization also heavily relies on the "Production Quality (PQ)" metric. PQ explicitly measures technical aspects like clarity, fidelity, dynamics, and frequency response, rather than subjective tastes. By optimizing for high fidelity and technical clarity, we primarily push the model towards realism and professional production standards, which are far more universal and less culturally specific than artistic or stylistic preferences. This anchors our "aesthetic" goal to a more objective standard, reducing the risk of encoding culturally-biased preferences.

  **(2) Scope Limitation:** The scope of our work is mainly focused on sound effects and instrumental music, deliberately excluding human speech and singing. This significantly reduces the risk of perpetuating some of the most harmful forms of representational bias related to accents, dialects, language, or vocal characteristics tied to specific demographic groups (e.g., gender, ethnicity). By avoiding human vocal content, we eliminate a major source of potential bias that could manifest through linguistic, accentual, or vocal timbre preferences. In future work, when incorporating human speech and singing, we will explore effective methods to mitigate these biases, such as diverse speaker representation, accent-inclusive training data, and bias-aware reward design.

  **(3) Diverse Rater Pool:** For the more subjective components of the Aesthetic Reward (such as Content Enjoyment), we deliberately chose the Meta Audiobox Aesthetics model. As emphasized in its technical report, this model was trained on ratings from 158 diverse raters from the general public, explicitly aiming to capture a broad spectrum of human judgments. The diversity of the rater pool helps ensure that the reward signal aggregates opinions across different cultural backgrounds, age groups, and personal preferences, rather than encoding a single, narrow cultural viewpoint. By leveraging a reward signal that already aggregates diverse opinions, we actively avoid encoding a single, narrow cultural viewpoint for the more subjective aspects of sound quality.

  While we do not claim to have eliminated all bias, we believe our specific methodological choices—from the scope of our task to the definition and measurement of "aesthetics"—serve as significant, practical mitigation steps. These choices reflect our commitment to responsible AI development and demonstrate that bias mitigation can be integrated into the core design of the system, rather than being treated as an afterthought. Future work should continue to prioritize inclusive dataset construction, culturally aware reward design, and ongoing evaluation of potential biases in generated content.

## I   POTENTIAL NEGATIVE SOCIETAL IMPACTS

While PrismAudio represents significant progress in video-to-audio generation technology, we acknowledge several potential negative societal impacts that warrant careful consideration and mitigation strategies.

- **Deepfake and Misinformation Risks.** The high-quality audio generation capabilities of PrismAudio could potentially be misused to create convincing fake audio content synchronized with video footage, contributing to the spread of misinformation or fabricated evidence. The multi-dimensional optimization across semantic, temporal, aesthetic, and

spatial dimensions makes generated audio particularly realistic, which could enhance the believability of manipulated media content. We strongly advocate for the development of corresponding detection technologies and recommend that generated content be clearly labeled as synthetic.

- **Creative Industry Displacement.** The sophisticated Chain-of-Thought reasoning and multi-dimensional quality optimization in PrismAudio may reduce demand for professional foley artists, sound designers, and audio post-production specialists. While this technology can democratize content creation and reduce production costs, it may also lead to job displacement in creative industries. We encourage the development of human-AI collaborative workflows that augment rather than replace human creativity and expertise.

We encourage responsible deployment of this technology with appropriate safeguards, transparent labeling of synthetic content, and continued research into bias mitigation and detection methodologies.

## J  THE USE OF LARGE LANGUAGE MODELS (LLMS)

In accordance with ICLR's guidelines, we disclose that LLMs were used during the preparation of this manuscript. We utilized LLMs exclusively as a writing aid to enhance the clarity and readability of the text.

The primary applications included:

- Proofreading for grammatical errors and typos.
- Rephrasing sentences for improved conciseness and flow.

All scientific contributions, including the core ideas, methodology, experimental design, and interpretation of results, are the original work of the human authors. The LLMs were not used for generating novel scientific content or analyses.

## K  SAFEGUARDS

We used a diverse training dataset covering a wide range of acoustic scenes to minimize reinforcing stereotypes or incorrect associations between sounds and specific demographic groups. The model will be released in stages to better assess its impact and improve safeguards. However, once the model is openly released, we cannot control how others use it. Therefore, we provide clear usage guidelines to encourage responsible use and help mitigate potential misuse.

