# OpenReview forum: "PrismAudio: Decomposed Chain-of-Thought and Multi-dimensional Rewards for Video-to-Audio Generation"
_ICLR.cc/2026/Conference — ICLR 2026 Poster_

### Official Review · Reviewer_oFRj · 2025-10-26

**Soundness:** 2
**Presentation:** 3
**Contribution:** 2
**Rating:** 4
**Confidence:** 4

**Summary:**

This paper proposes PrismAudio, a video‑to‑audio (V2A) generation framework that uses multi‑dimensional Chain‑of‑Thought (CoT) (Semantic, Temporal, Aesthetic, and Spatial) with reinforcement learning.

To build PrismAudio, the authors
- replace CLIP and T5 with VideoPrism and T5‑Gemma, respectively, on top of the model architecture of ThinkSound
- extend ThinkSound’s monolithic CoT planning into the four CoTs
- introduce Fast‑GRPO, a hybrid ODE–SDE variant of Group Relative Policy Optimization tailored to flow‑matching models for efficient optimization.

The paper also presents AudioCanvas, a new benchmark of 3,177 real‑world videos that covers 300 single‑event classes and 501 multi‑event samples, curated with high alignment and structured CoT annotations by using Gemini 2.5 Pro from AudioSet.

**Strengths:**

- Building on ThinkSound’s finding that CoT planning can help V2A, the paper proposes the efficient training framework of multi‑dimensional CoT and shows that further improves performance across semantic, temporal, aesthetic, and spatial axes—without sacrificing any one axis, according to both objective metrics and MOS on VGGSound and AudioCanvas dataset.
- The paper includes comprehensive ablations (encoder choices, CoT structure, reward composition, Fast‑GRPO efficiency) on both objective and subjective evaluations, which supports the effectiveness of proposed framework.

**Weaknesses:**

- Problem setup is still “CoT‑based V2A” in the ThinkSound lineage; the paper’s main conceptual step is decomposition from a monolithic to a multi‑dimensional CoT with reward alignment. This is a natural extension that is practically useful but arguably incremental. Note that  the reviewer acknowledges its effectiveness.

- Although the reviewer acknowledges that the random window approach is effective for applying SDE steps to efficiently optimize flow matching, the significance of this contribution remains limited. In the context of diffusion models [1] and its fundamental equivalence with flow matching [2,3], the known relationship between the SDE and the PF-ODE [4] already makes it theoretically clear that one can switch between ODE and SDE steps during the sampling process. Consequently, Fast-GRPO demonstrates an empirical gain achieved through a heuristic technique rather than presenting a novel fundamental contribution.

- Dataset construction details leave critical ambiguities.
    - VGGSound test split / inference‑time CoTs: At inference, PrismAudio appears to require a structured CoT. It is unclear how these CoTs are obtained on VGGSound test videos—e.g., are they generated by the fine‑tuned VideoLLaMA2 from the test videos alone, or are they derived from the original tag-level captions? The Implementation section can be interpreted that the authors annotate VGGSound testsplit with CoTs using fine‑tuned VideoLLaMA2, but does not say whether "sounding" videos or "silent" videos are input to that model to get CoTs, which the reviewer the silent ones are used. If this is the case, is a gap observed between the training and inference setups? This is because during training, the model uses VideoLLaMA2 outputs obtained from "sounding" videos, whereas at inference, it would be using outputs obtained from "silent" videos, creating a gap.
   - Regarding the constructing AudioCanvas dataset, are "sounding" videos used as the input or are "silent" videos fed into Gemini 2.5 Pro? If the CoTs are generated from "sounding" videos, what would PrismAudio's performance be when evaluated using CoTs generated from "silent" videos as input?

[minor]
- The paper seems to use single‑stimulus subjective evaluation. However, MUSHRA (multi‑stimulus) or pairwise preference tests (e.g., [5,6]) would provide more discriminative evidence if authors want to claim the effectiveness of proposed methods based on human evaluation.

[1] Karras, Tero, et al. "Analyzing and improving the training dynamics of diffusion models." CVPR 2024.

[2] Lu, Cheng, et al. "Simplifying, stabilizing and scaling continuous-time consistency models." ICLR 2025.

[3] https://diffusionflow.github.io/

[4] Song, Yang, et al. "Score-based generative modeling through stochastic differential equations."  ICLR 2021.

[5] Evans, Zach, et al. "Fast Timing-Conditioned Latent Audio Diffusion." ICML 2024

[6] Novak, Zachary, et al. "Presto! Distilling Steps and Layers for Accelerating Music Generation." ICLR 2025

**Questions:**

- What are the exact prompts to Gemini 2.5 Pro to construct dataset for both VideoLLaMA2 and AudioCanvas?
- Could you explain the training details of VideoLLaMA2?

See weaknesses as well.

---

> ### Author Response · Authors · 2025-11-21
> **Response to Reviewer oFRj [1/3]**
>
> We sincerely thank Reviewer oFRj for your thoughtful review and constructive feedback. **Please first check our comment under "Global Response to All Reviewers and ACs"**, which addresses all questions and concerns shared by multiple reviewers. We hope the following point-by-point response will resolve all your remaining questions and concerns.
>
> ---
> > **Q1:** The decomposition from monolithic to multi-dimensional CoT is a natural extension that is practically useful but arguably incremental. Note that the reviewer acknowledges its effectiveness.
>
> We thank the reviewer for acknowledging the effectiveness of PrismAudio. As stated in our main contributions, **PrismAudio is the first V2A framework to integrate specialized CoT modules with multi-dimensional RL optimization**, providing a systemic solution to a **fundamental yet previously unsolved problem in V2A: Objective Entanglement.**
>
> 1. **Conceptual Contribution to Address Objective Entanglement**
>
> As recognized by Reviewer 2VpK (`the paper clearly defines the problem as objective entanglement`) and Reviewer BAuq (`the four decomposition dimensions (semantic, temporal, aesthetic, and spatial) are conceptually sound and well-motivated for the V2A task`), PrismAudio addresses the fundamental limitation of current V2A approaches in handling competing goals (semantic, temporal, aesthetic, spatial). ThinkSound, despite pioneering CoT, fails to solve this problem due to the following limitations:
>
> *   **Monolithic Planning:** ThinkSound conflates distinct analytical tasks into a single path, leading to **inadequate treatment** **of each dimension** and **multimodal hallucinations** in complex scenarios.
>
> *   **Objective Entanglement:** A unified reconstruction loss conflates competing perceptual goals, failing to learn necessary **context-dependent trade-offs**.
>
> *   **Absence of Preference Alignment:** ThinkSound lacks **RL-based feedback** to optimize for human preferences, producing **technically correct but perceptually unsatisfying results**.
>
>
> To tackle these limitations, **PrismAudio unlocks the full potential of CoT in generative tasks by coupling it with robust reward functions, transforming complex V2A reasoning from an open-ended guidance into an optimizable objective**.
>
> **2. Systematic Innovation: CoT-RL Integration** This transformation is achieved through four synergistic innovations:
>
> *   **Decoupled CoT Construction:** We introduce four axis-specific CoT modules (Section 3.2), which are not merely for interpretability, but **to explicitly expose distinct reasoning paths** that can be targeted by specific reward functions.
>
> *   **Aligned Reward Functions:** Each CoT module is paired with a targeted reward function, enabling **precise, axis-wise feedback** (Section 3.3.1) that was impossible in unified loss settings.
>
> *   **Multi-dimensional GRPO:** A principled RL algorithm using group-normalized rewards to ensure **balanced, simultaneous optimization** **across all dimensions** (Sec 3.3.2).
>
> *   **Enhanced Architecture:** Upgrades to T5-Gemma and VideoPrism specifically enhance reasoning capabilities to handle complex video scenarios required for multi-dimensional planning (Appendix E.1).
>
>
> **3. Empirical Evidence of Effectiveness** Experimental results demonstrate that this integrated framework solves problems that CoT alone (ThinkSound) cannot:
>
> *   **Quantitative Results (SOTA & Solving Trade-offs):** PrismAudio achieves **SOTA performance across all perceptual axes** on both VGGSound and AudioCanvas (Tables 1 & 2). **Reviewer 2VpK** confirms that the model "achieves high audio quality and SOTA balance across all axes... with faster inference and strong robustness." Furthermore, our ablation study (**Table 4**) empirically proves we resolved objective entanglement: optimizing single dimensions degrades others, while our framework achieves simultaneous improvement, acknowledged by **Reviewer BAuq**.
>
> *   **Qualitative Results (Mitigating Hallucinations and Better Aesthetic Quality):** **PrismAudio substantially mitigates hallucination issues** (spurious voices, background music) observed in ThinkSound and MMAudio. As demonstrated in the _"Comparison with Baselines in Multi-Event Scenarios on the AudioCanvas Benchmark"_ section of our project page, **PrismAudio generates clean, contextually appropriate audio for complex scenarios** like "Chopping, Thudding" and "Machining, Tapping" while baseline methods produce unexpected artifacts. Additionally, music examples on the project page demonstrate PrismAudio's superior audio quality with enhanced fidelity, naturalness, and audio-visual synchronization compared to previous SOTA methods.

---

> ### Author Response · Authors · 2025-11-21
> **Response to Reviewer oFRj [2/3]**
>
> > **Q2:** Fast-GRPO demonstrates an empirical gain through a heuristic technique rather than presenting a novel fundamental contribution, as the SDE-ODE equivalence is theoretically known.
>
> We acknowledge that SDE-ODE equivalence is well-established[1]. In fact, the SDE-ODE equivalence is the foundational principle that enables GRPO to be applied to flow-matching models in the first place, as seen in prior work such as Flow-GRPO[2] and DanceGRPO[3]. This equivalence makes GRPO application to flow matching **possible**, but **the computational tractability challenge remains to be addressed**. Prior work such as Flow-GRPO requires SDE sampling at every step, creating substantial training overhead that makes multi-dimensional RL optimization computationally intractable.
>
> The **key contribution of Fast-GRPO** lies in **making multi-dimensional GRPO practically feasible for flow-matching models through a carefully designed hybrid sampling and optimization pipeline**, as acknowledged by **Reviewer 2VpK**. Specifically, we introduce three tightly integrated components:
>
> 1.  **Reward Computation Adapted to Hybrid Sampling (Section 3.3.2):** Given our windowed hybrid ODE–SDE sampling—where stochasticity is limited to a small time window—**we reformulate the GRPO objective to operate only on SDE steps**, where valid probability densities exist. This design avoids ill-defined likelihood ratios over deterministic ODE segments and enables stable, tractable policy gradients. As detailed in Section 3.3.2, the resulting multi-dimensional GRPO objective aligns reward feedback with the stochastic part of the trajectory, achieving both computational efficiency and compatibility with flow-matching dynamics.
>
> 2.  **Closed-form Policy Ratio (Appendix B.1):** By limiting SDE steps to a window $w \ll T$, we enable closed-form policy ratio computation, **reducing complexity from O(T) to O(w)**.  **Reviewer 2VpK** recognizes that our closed-form policy ratio computation enables "near-linear GRPO training complexity." **Critically, this complexity reduction represents the key bridge from "theoretical possibility" to "practical feasibility"**: while SDE-ODE equivalence makes GRPO theoretically applicable to flow matching, the O(T) complexity of full SDE sampling renders multi-dimensional RL optimization computationally intractable in practice; in contrast, our O(w) solution (with w=2 << T=50) is the critical enabler that facilitates practical and scalable multi-dimensional RL.
>
> 3.  **Stable Multi-Dimensional Policy Optimization (Section 3.3.2, Appendix B.3):** We use a hybrid sampling scheme that, at each iteration, places a small stochastic window at a random position along the full trajectory—using SDE only within the window and ODE elsewhere. Over training, **this ensures every segment can be stochastic, preventing systematic bias towards fully deterministic or stochastic paths**. Combined with ratio clipping, group-relative normalization, and a KL penalty (λ=0.04) to curb reward hacking—as recognized by **Reviewer 2VpK**—Fast-GRPO enables robust, efficient multi-dimensional GRPO optimization.
>
>
> Empirically, **Fast-GRPO achieves 1.6× faster convergence than Flow-GRPO** (200 vs. 600+ steps) **while achieving higher final reward** (0.51 vs. 0.47 CLAP, Section 4.3, Figure 4). The complexity reduction from O(T) to O(w) with Fast-GRPO enables the comprehensive experimental validation that **Reviewer** **2VpK and** **BAuq** acknowledge.
>
> ---
>
> > **Q3:** What are the exact prompts to Gemini 2.5 Pro for constructing datasets (VideoLLaMA2 and AudioCanvas)? Could you explain the training details of VideoLLaMA2?
>
> We have incorporated these implementation details in the revised manuscript (Appendix D.2 and D.3).
>
> 1.  **Gemini 2.5 Pro Prompt:** Both VideoLLaMA2 training data and AudioCanvas use the same prompt (Appendix D.2), which instructs the model to generate structured CoT descriptions covering four dimensions: semantic, temporal, aesthetic, and spatial. PrismAudio then uses a text LLM to transform these captions into our desired multi-dimensional decoupled CoT input format (the transformation prompt is also provided in Appendix D.2).
>
> 2.  **VideoLLaMA2 Training Details:** We use the official VideoLLaMA2 repository with DeepSpeed ZeRO-3, initializing from VideoLLaMA2-AV (7B). We employ AdamW optimizer (lr=2e-5), cosine annealing with warmup, batch size 4 per GPU (global 128), and train for 10 epochs. We freeze the video encoder, audio encoder, and audio projector; only the video projector and LLM are trainable. This design helps bridge the gap between training (sounding videos) and inference (silent videos). Complete details are provided in Appendix D.3.

---

> ### Author Response · Authors · 2025-11-21
> **Response to Reviewer oFRj [3/3]**
>
> > **Q4:** Critical ambiguities regarding whether "sounding" or "silent" videos are used for CoT generation during training vs. inference, potentially creating a train-test gap.
>
> Thank you for the valuable question. We apologize for the ambiguity in our paper. We clarify that the CoT generation process reflects deliberate design choices for different evaluation scenarios in the **VT2A (Video-Text-to-Audio)** setting. The details are as follows:
>
> For **AudioCanvas**, we use **sounding videos** as input to Gemini 2.5 Pro to construct high-quality and **oracle ground truth** CoT captions for VT2A evaluation. As validated in Appendix C.2, the annotations achieve **96.5% semantic correctness** and **94.2% temporal accuracy**, quantitatively validating the high quality of AudioCanvas. To ensure fair comparison, **all baselines and PrismAudio use the same CoT captions for evaluation**. The difference is that PrismAudio uses a text LLM to transform these captions into our desired multi-dimensional decoupled CoT input format (Appendix D.2).
>
> For **VGGSound**, to ensure fair comparison with other baselines on this established benchmark, we use **silent videos** (audio removed) combined with tag-level text captions to generate CoTs during inference, using our fine-tuned VideoLLaMA2 model. This setup matches the standard VT2A evaluation protocol, where models receive video and text inputs, but the video component is silent (audio removed) to simulate real-world deployment scenarios. We have included detailed implementation details in Appendix D.
>
> **Additional Experiment to Address the Concern on Train-Test Gap:**
>
> To address the reviewer's concern about the train-test gap, we conducted an additional experiment using **silent videos** as input to Gemini 2.5 Pro for CoT generation on AudioCanvas (results shown in Table 2, highlighted in blue). As expected, **the absence of audio information in CoT generation does impact performance** compared to using sounding videos. However, **PrismAudio remains effective with visual-only CoT**, **achieving** **strong performance gains over ThinkSound in all critical dimensions** (Temporal DeSync: 0.42 vs. ThinkSound's 0.80, Aesthetic Quality PQ: 6.55 vs. ThinkSound's 6.48, Spatial Accuracy CRW: 14.26 vs. ThinkSound's 22.82). These results demonstrate that **our multi-dimensional CoT-RL framework is robust to CoT quality variations**. We will release both versions of CoT captions (generated from sounding videos and silent videos) to accommodate different comparison scenarios.
>
> ---
>
> > **Q5:** The paper uses single-stimulus evaluation, but MUSHRA or pairwise preference tests would provide more discriminative evidence.
>
> We appreciate the Reviewer's valuable suggestion. In response to this feedback, we conducted additional blind pairwise preference tests comparing PrismAudio against ThinkSound and MMAudio on 200 videos from VGGSound and 200 from AudioCanvas, with 10 expert annotators per pair across four dimensions. **PrismAudio vs. ThinkSound**: PrismAudio is preferred in **76.2%** (semantic), **72.7%** (temporal), **83.9%** (aesthetic), and **77.4%** (spatial) of comparisons. **PrismAudio vs. MMAudio**: PrismAudio is preferred in **78.5%** (semantic), **70.3%** (temporal), and **85.1%** (aesthetic) of comparisons (Note that MMAudio generates monaural audio). **These pairwise preference tests provide more discriminative evidence than single-stimulus evaluation, and the results are consistent with our MOS evaluations (Tables 1 and 2), consistently validating PrismAudio's superior performance across all perceptual dimensions.**
>
> We also encourage the reviewer to experience the audio quality directly through our project page (shown in the last sentence of Abstract), which provides extensive side-by-side comparisons across in-domain (VGGSound) and out-of-domain (AudioCanvas) evaluations, single-event and multi-event scenarios, and audio generation for Sora2 and Veo3 generated videos.
>
> ---
>
> ## References
>
> [1] Song, Yang, et al. "Score-based generative modeling through stochastic differential equations." ICLR 2021.
>
> [2] Liu J, Liu G, Liang J, et al. Flow-grpo: Training flow matching models via online rl[J]. arXiv preprint arXiv:2505.05470, 2025.
>
> [3] Xue, Zeyue, et al. "DanceGRPO: Unleashing GRPO on Visual Generation." arXiv preprint arXiv:2505.07818 (2025).

---

> > ### Comment · Reviewer_oFRj · 2025-11-26
> >
> > The reviewer appreciates the authors' detailed explanations and the additional experiments and ablations.
> >
> > One of the major concerns was, for CoTs creation during inferece, whether the proposed method required sounding videos to achieve sufficient performance or not. This was a critical point because such a dependency would deviate from the real-world use case of generating sound for silent videos.
> >
> > However, the authors clarify that silent videos are used for VGGSound experiments and, even on AudioCanvas dataset, the proposed framework shows the performance gain with CoTs from silent videos compared with baselines. This addresses the major concern.
> >
> > Thus, the reviewer has decided to raise the score. Thank you very much for your thorough replies.

---

> > > ### Author Response · Authors · 2025-11-26
> > > **Thank you for the positive feedback and raising the score**
> > >
> > > We sincerely thank the reviewer for confirming that our responses have effectively addressed your concerns, and for the decision to raise the score!
> > >
> > > We are glad to hear that our clarification regarding the method's effectiveness on silent videos resolved your core concern. We agree that this distinction is crucial for demonstrating the practical value of our research.
> > >
> > > Thanks again for your time and valuable reviews.

---

### Official Review · Reviewer_BAuq · 2025-10-31

**Soundness:** 4
**Presentation:** 3
**Contribution:** 4
**Rating:** 8
**Confidence:** 3

**Summary:**

The paper proposes PrismAudio, a video-to-audio (V2A) generation framework that decomposes ThinkSound planning into four dimensions (semantic, temporal, aesthetic, and spatial reasoning).
Each dimension is associated with a representative reward, and the model is trained via reinforcement learning to balance these objectives.
They also provide Fast-GRPO which utilizes both SDE and ODE at the same in order to enable the efficient optimization.
With a smaller number of parameters and shorter inference time, PrismAudio achieves state-of-the-art performance on both VGGSound (in-domain) and AudioCanvas (out-of-domain) benchmarks across all four evaluation dimensions.

**Strengths:**

1.	The proposed AudioCanvas Benchmark is valuable and provides high-quality, well-structured evaluation data.
2.	The four decomposition dimensions (semantic, temporal, aesthetic, and spatial) are conceptually sound and well-motivated for the V2A task.
3.	Extensive experiments and ablation analyses comprehensively support the model’s effectiveness and robustness across different dimensions.

**Weaknesses:**

1. The training pipeline is complex, involving multiple reward signals from different models, which may limit reproducibility and dependency problem.
2. In Table 4, the ablation study evaluates single-dimensional rewards, but it remains unclear whether all four dimensions (semantic, temporal, aesthetic, spatial) are truly necessary. In many video-to-audio scenarios, only semantic and temporal aspects may suffice, while spatial or aesthetic factors could be redundant. Demonstrating or analyzing such reduced configurations could clarify whether the full four-dimensional setup is essential or unnecessarily complex.

**Questions:**

Given that training the VAE requires 24 GPUs, have the authors considered excluding VAE fine-tuning and the spatial reward?

---

> ### Author Response · Authors · 2025-11-21
> **Response to Reviewer BAuq**
>
> We sincerely thank Reviewer BAuq for your positive reviews and constructive feedback. **Please first check our comment under "Global Response to All Reviewers and ACs"**, which addresses all questions and concerns shared by multiple reviewers. We hope the following point-by-point response will resolve all your remaining questions and concerns.
>
> ---
> > **Q1:** Necessity of All Four Dimensions: Are semantic and temporal aspects sufficient, or are spatial and aesthetic factors necessary?
>
> We thank the reviewer for this insightful suggestion.
>
> **Quantitative Evidence**:
>
> To rigorously validate the necessity of our full four-dimensional framework, we conducted additional ablation studies on the **VGGSound test set**, evaluating three reduced configurations: (1) removing Aesthetic reward (Sem+Temp+Spatial), (2) removing Spatial reward (Sem+Temp+Aesthetic), and (3) removing both Aesthetic and Spatial rewards (Sem+Temp only). Detailed quantitative results are presented in Appendix E.7 (Table 13).
>
> On the VGGSound test set, reduced configurations suffer from significant degradation in critical dimensions. The **Sem+Temp only** configuration shows no improvement in spatial accuracy (CRW: 8.29 → 8.38) and only marginal aesthetic gains (PQ: 6.24 → 6.30), while FD degrades (1.10 → 1.22), indicating lower audio quality. **Adding only Spatial reward** improves spatial metrics (CRW: 8.38 → 7.53) but fails to enhance aesthetics (PQ: 6.30 → 6.19). **Adding only Aesthetic reward** improves aesthetics (PQ: 6.30 → 6.44) but degrades audio quality (FD: 1.22 → 1.52). In contrast, our **full four-dimensional approach** is the only configuration achieving **holistic improvement across all axes simultaneously** over Baseline (No RL) (CLAP: 0.44 → 0.47, DeSync: 0.48 → 0.41, PQ: 6.24 → 6.38, CRW: 8.29 → 7.72, FD: 1.10 → 1.08). These results confirm that **aesthetic qualities and spatial accuracies are not "free" benefits from semantic/temporal optimization but distinct objectives requiring explicit optimization**.
>
> **Qualitative Evidence:**
>
> Our project page (the last sentence of Abstract) provides compelling qualitative evidence demonstrating the necessity of all four dimensions. **Note that the baseline methods (ThinkSound, MMAudio) suffer from objective entanglement and also do not optimize on aesthetic and spatial dimensions.** (1) In music scenarios, baseline methods (ThinkSound, MMAudio) generate audio that lacks naturalness and contextual appropriateness, while **PrismAudio with explicit aesthetic modeling generates more authentic and perceptually rich audio.** (2) In scenarios with clear spatial motion (e.g., **"Accelerating, Revving, Vroom"**), baseline methods fail to maintain proper spatial localization, producing inconsistent left-right channel relationships, while **PrismAudio accurately captures spatial dynamics.** (3) In multi-event scenarios (e.g., **"Chopping, Thudding", "Ice Cutting, Freezer Shutting"**), baseline methods frequently generate spurious artifacts such as unexpected human voices and background music unrelated to visual content, indicating objective entanglement, while **PrismAudio with explicit objective disentanglement generates smooth, natural, and contextually appropriate audio without such hallucinations**.
>
> ---
> > **Q2:** Given that training the VAE requires 24 GPUs, have the authors considered excluding VAE fine-tuning and the spatial reward?
>
> **Regarding VAE Fine-tuning:**
>
> We acknowledge that VAE fine-tuning requires substantial computational resources (24 A800, ~5 days). However, **VAE fine-tuning is optional**. We have added ablation studies in **Table 10 of the updated manuscript**, which demonstrate that **fine-tuning can improve VAE reconstruction capability, but the improvement is limited because the pre-trained VAE already provides good reconstruction quality**. For researchers with limited computational resources, we will provide the pre-trained VAE checkpoints that we use in this work, which can be directly employed without fine-tuning.
>
> **Regarding Spatial Reward:**
>
> **The spatial reward (StereoCRW) is essential for our task, which aims to generate stereo audio.** As shown above, removing the spatial reward leads to substantial degradation in spatial accuracy (CRW: 7.72 → 7.84) and overall audio quality (FD: 1.08 → 1.52). Moreover, previous works such as ThinkSound and AudioX fail to maintain proper left-right channel consistency compared to ground truth (Tables 1 and 2). The spatial dimension is critical for accurate stereo sound generation. To accommodate researchers with different computational resources, we will also release a model variant without the spatial reward.
>
> ---
>
> > **Q3:** The training pipeline is complex, involving multiple reward signals from different models, which may limit reproducibility and dependency problems.
>
> Please refer to **Global Response - Q2: Reproducibility and Open-Source Commitment** for our detailed response.

---

> ### Author Response · Authors · 2025-11-27
> **Gentle Reminder Regarding Our Response**
>
> We sincerely appreciate your strong support and the thoughtful insights you have dedicated to our work. Your constructive feedback has been invaluable to us.
>
> We have carefully addressed your concerns point-by-point in our rebuttal. Please feel free to let us know if you have any further questions or require additional clarifications.
>
> Thank you again for your time and positive evaluation.

---

> > ### Comment · Reviewer_BAuq · 2025-11-27
> >
> > I appreciate that the authors addressed my concerns with thorough experiments. It is an interesting finding that using only the semantic and temporal rewards increases the CLAP score. In addition, the spatial metrics and distribution quality (FD) reach their best values (excluding the aesthetic metric) when the aesthetic reward is removed. Likewise, the aesthetic score is highest when the spatial reward is excluded.
> >
> > Therefore, I do not fully agree that a multi-dimensional reward system is essential based on this result. These observations support adopting a more flexible approach to reward design rather than treating all reward components as essential. For relatively simple 9-second audio samples, it may be more informative to highlight the trade-offs across reward configurations and to clarify which checkpoints users may prefer depending on their priorities (e.g., some users may not require stereo sound at all).

---

> > > ### Author Response · Authors · 2025-11-27
> > > **Follow-up Response to Reviewer BAuq**
> > >
> > > Thank you very much for the thoughtful follow-up and your keen analysis of the trade-offs in our ablation results. We genuinely appreciate these detailed observations.
> > >
> > > **Regarding the trade-offs**: Your observation precisely illustrates the challenge of **objective entanglement** discussed in our paper. As you noted, removing one constraint (like the spatial reward) can indeed slightly relieve the optimization burden on other dimensions (like aesthetic quality). However, this comes at the cost of losing control over the removed dimension. PrismAudio is specifically designed to mitigate this conflict, finding the best possible equilibrium where all dimensions perform well simultaneously.
> > >
> > > **On flexibility for users**: We **strongly agree your suggestion** that `it may be more informative to highlight the trade-offs across reward configurations and to clarify which checkpoints users may prefer depending on their priorities (e.g., some users may not require stereo sound at all)`. We agree that distinguishing between "essential" and "optional" components depends on the user's specific goal (e.g., generating monaural sound vs. immersive spatial audio). Therefore, consistent with our **Reproducibility Statement**, we commit to **open-sourcing multiple checkpoints with different reward configurations along with the full training code**. This allows users to choose the version that best matches their priorities, significantly enhancing the practical value of our work.
> > >
> > > **Why the full model matters**: That said, we maintain that the full PrismAudio configuration is "essential" for **comprehensive high-fidelity generation**. As shown in **Table 4 and Table 13**, it is the only configuration that achieves substantial improvements across all axes simultaneously. For applications demanding the highest level of realism—where semantic, temporal, spatial, and aesthetic qualities are all non-negotiable—the complete PrismAudio model remains the optimal choice.
> > >
> > > Thank you again for your constructive feedback, which helps us better communicate the trade-offs and practical applicability of our framework.

---

### Official Review · Reviewer_2VpK · 2025-10-31

**Soundness:** 4
**Presentation:** 4
**Contribution:** 3
**Rating:** 6
**Confidence:** 3

**Summary:**

This paper proposes PrismAudio for video-to-audio generation. The core challenge is objective entanglement across four perceptual goals which are semantic consistency, temporal synchrony, aesthetic quality, and spatial accuracy. PrismAudio addresses this by decomposing  reasoning into four Chain-of-Thought (CoT) modules,  each paired with a targeted reward.  The base generator is a Diffusion Transformer with flow matching, conditioned on axis-specific reasoning text and post-trained with Group Relative Policy Optimization (GRPO) using group-normalized, axis-wise rewards. To reduce overhead, Fast-GRPO interleaves brief SDE windows into an otherwise ODE trajectory. This preserves the terminal distribution yet maintains exploration and reduces cost. The paper reports state-of-the-art results across all four axes on VGGSound (in-domain) and AudioCanvas (out-of-domain), with faster inference than prior SOTAs.

**Strengths:**

The paper clearly defines the problem as objective entanglement and addresses it by decomposing reasoning into four Chain-of-Thought (CoT) axes with aligned, axis-wise rewards under reinforcement learning. Fast-GRPO limits stochastic exploration to a short, randomly placed window along the trajectory using a mixed ODE–SDE sampler, which enables closed-form per-step policy ratios and stable group-relative updates via weighted aggregation, within-group normalization, and ratio clipping, yielding near-linear GRPO training complexity (reducing policy-model NFE from T to w) without compromising reward computation. The authors also acknowledge reward hacking risks and add a KL penalty (λ=0.04) for stability. Empirically, the model achieves high audio quality and SOTA balance across semantic, temporal, aesthetic, and spatial axes on VGGSound and AudioCanvas, with faster inference and strong robustness. Audio generation results are released upon the public github. But there is no code.

**Weaknesses:**

	The method explores with SDE only within a randomly placed window and uses ODE elsewhere. If that window misses moments where scene dynamics are most critical for sound perception, those segments may be under-learned. This raises questions about coverage and stability under the proposed sampling scheme, and whether Fast-GRPO still acquires the correct behavior when the window skips the most eventful frames.

1. The paper argues that mixing ODE and SDE preserves the terminal distribution. As parameters are updated and numerical errors accrue during training, it would be helpful to show that this property and the resulting reward estimates remain stable.

2. If the policy inside the window is modeled as a fixed-form Gaussian, it may be too rigid to capture anisotropic or structured uncertainty in complex latent spaces, potentially limiting performance.

3. Although inference is lightweight, the training phase appears to require substantial compute and memory.

**Questions:**

Following the weaknesses above, here are several questions for the authors:
1. If the random SDE window skips the most eventful frames (e.g., sudden scene or sound changes), does Fast-GRPO still converge to the intended behavior? Do you report coverage statistics or guarantees over training?
2. Under parameter updates during training, does mixed ODE–SDE sampling approximately preserve the terminal distribution? How sensitive are reward estimates and learning signals to accumulated numerical error or parameter drift?
3. Is the in-window policy fixed as an isotropic Gaussian? If so, did you try diagonal/low-rank/mixture variants?
4. For inference, what GPU resources do you require, specifically VRAM usage?
5. On the VGGSound dataset, how many test videos did you evaluate (exact count for your test split)?

**Details Of Ethics Concerns:**

The paper explicitly discusses the following potential negative impacts about the training data and reward functions, especially the Aesthetic CoT module and its reward signals, may inadvertently encode cultural bias and representation issues regarding what counts as “good sound” or appropriate audio-visual relationships.
However, given that the data were collected from publicly available datasets and the intended use is academic and research only, the authors contend that there are no material ethics concerns.

---

> ### Author Response · Authors · 2025-11-21
> **Response to Reviewer 2VpK [1/3]**
>
> We sincerely thank Reviewer 2VpK for your positive reviews and valuable feedback. **Please first check our comment under "Global Response to All Reviewers and ACs"**, which addresses all questions and concerns shared by multiple reviewers. We hope the following point-by-point response will resolve all your remaining questions and concerns.
>
> ---
> > **Q1:** If the random SDE window skips the most eventful frames (e.g., sudden scene or sound changes), does Fast-GRPO still converge to the intended behavior?
>
> We apologize for the ambiguity in our paper that led to this confusion. In Section 3.3.2, Equations 2, 3, and 4, the variable $t$ refers to the **diffusion noise schedule (timesteps)**, not the temporal duration of the audio. We clarify this crucial distinction below.
>
> As shown in Equation 2 of our paper, the update rule is:
>
> $ \mathbf{x}\_{t+1} = \begin{cases} \mathbf{x}\_t + v\_\theta(\mathbf{x}\_t, t, c)\Delta t, & \text{if } t \notin \mathcal{W}(\ell) \quad \text{(ODE step)} \\\\ \mathbf{x}\_t + \mu\_{\text{SDE}}(\mathbf{x}\_t, t, c)\Delta\_t + \sigma\_t\sqrt{\Delta\_t}\ \varepsilon\_t, & \text{if } t \in \mathcal{W}(\ell) \quad \text{(SDE step)} \end{cases}$
>
> Here, $t$ indexes **diffusion timesteps** in the noise schedule, and $\mathbf{x}_t \in \mathbb{R}^{B \times C \times L}$ represents the **entire audio latent sequence** at diffusion timestep $t$. The window $\mathcal{W}(\ell)$ determines _when_ in the denoising process (which diffusion timestep) we apply stochastic exploration, not _where_ in the audio. At each diffusion step, both ODE and SDE solvers operate on the entire latent simultaneously, with noise $\varepsilon_t$ affecting all temporal frames uniformly. Therefore, the random SDE window will **not skip eventful moments** in the audio temporal dimension, and Fast-GRPO converges to the intended behavior across the entire audio duration.
>
> ---
> > **Q2:** Under parameter updates during training, does mixed ODE–SDE sampling approximately preserve the terminal distribution?
>
> **Yes.** Mixed ODE–SDE sampling approximately preserves the terminal distribution under parameter updates during training[1]. The key insight is that, provided that the model parameters remain fixed during a single sampling pass and the flow-matching model ideally learns the true underlying vector field, the theoretical equivalence between ODE and SDE formulations (illustrated in Sections B.1–B.2) **guarantees identical marginal distributions at all times, including the terminal distribution**.
>
> **Theoretical Foundation:**
>
> The preservation property follows the well-established equivalence between SDE and ODE formulations. Consider the SDE:
>
> $ d\mathbf{x} = \mathbf{f}(\mathbf{x}, t)dt + \mathbf{G}(\mathbf{x}, t)d\mathbf{w} $
>
> The marginal density $p_t(\mathbf{x})$ evolves according to the Fokker-Planck equation [2]:
>
> $ \frac{\partial p_t(\mathbf{x})}{\partial t} = -\sum_{i=1}^{d} \frac{\partial}{\partial x_i} [f_i(\mathbf{x}, t)p_t(\mathbf{x})] + \frac{1}{2} \sum_{i=1}^{d} \sum_{j=1}^{d} \frac{\partial^2}{\partial x_i \partial x_j} \left[\sum_{k=1}^{d} G_{ik}(\mathbf{x}, t)G_{jk}(\mathbf{x}, t)p_t(\mathbf{x})\right] $
>
> Using the product rule, this can be rewritten as:
>
> $ \frac{\partial p_t(\mathbf{x})}{\partial t} = -\sum_{i=1}^{d} \frac{\partial}{\partial x_i}[\tilde{f}_i(\mathbf{x}, t)p_t(\mathbf{x})] $
>
> where $\tilde{\mathbf{f}}(\mathbf{x}, t) := \mathbf{f}(\mathbf{x}, t) - \frac{1}{2}\nabla \cdot [\mathbf{G}(\mathbf{x}, t)\mathbf{G}(\mathbf{x}, t)^T] - \frac{1}{2}\mathbf{G}(\mathbf{x}, t)\mathbf{G}(\mathbf{x}, t)^T\nabla_{\mathbf{x}} \log p_t(\mathbf{x})$.
>
> This is precisely the continuity equation for the ODE $d\mathbf{x} = \tilde{\mathbf{f}}(\mathbf{x}, t)dt$. **Therefore, the SDE and its corresponding ODE induce identical marginal distributions** $p_t(\mathbf{x})$ **at all times**, including the terminal distribution.
>
> **Empirical Validation:**
>
> To **empirically validate the practical effectiveness of this theoretical property** and assess the impact of model approximation errors, we track representative samples throughout the training process and visualize how ODE and SDE distributions evolve across training steps (**see Appendix E.6 and Figure 5 for details**). In our visualization, star markers represent ODE-generated results, while circles of the same color represent SDE-generated results from the same input. As observed in the figure, as training advances, **the ODE and SDE distributions remain closely aligned**, demonstrating that despite the finite model capacity, **the distribution equivalence holds with high fidelity in practice**. This confirms that **mixed ODE–SDE sampling approximately preserves the terminal distribution even under iterative parameter updates during training**.

---

> ### Author Response · Authors · 2025-11-21
> **Response to Reviewer 2VpK [2/3]**
>
> > **Q3:** Sensitivity of Reward Estimates and Learning Signals
>
> Although discretization and numerical errors introduce small deviations in practice, the sensitivity of reward estimates and learning signals to these errors is well-controlled through multiple mechanisms:
>
> **1. Self-Correcting Nature through Flow Matching's Predictor-Corrector Property:**
>
> Flow matching naturally mitigates accumulated errors because at each step, the model predicts $v_\theta(\mathbf{x}_t, t, c)$ from the **current state** $\mathbf{x}_t$, not from an assumed ideal trajectory. If numerical errors cause deviation, the next velocity prediction automatically accounts for the actual (perturbed) state, steering it back toward the target distribution. This self-correcting mechanism prevents error accumulation over the sampling process.
>
> **2. Robustness via VAE Manifold Constraint and GRPO Filtering:**
>
> Our approach performs diffusion in the VAE-compressed latent space, which dramatically decreases the number of floating-point operations per step and thus reduces numerical error accumulation proportionally. Moreover, even if significant (albeit rare) numerical errors or exploratory samples result in verifiably poor-quality outputs, they are naturally suppressed by the GRPO mechanism: such samples receive lower reward scores and are assigned negative weights during optimization, effectively preventing the model from reinforcing these undesirable deviations.
>
> **3. Stabilized Learning Signals through Algorithmic Safeguards:**
>
> Even when substantial reward deviations occur, our learning signals remain stable due to three key algorithmic safeguards:
>
> *   **Within-group reward normalization** prevents outlier rewards from dominating gradient updates
>
> *   **Ratio clipping in the GRPO objective** bounds the magnitude of policy updates
>
> *   **Gradient norm clipping** prevents exploding gradients
>
>
> **Empirical Validation of Reward Robustness:**
>
> To validate the robustness of our reward estimation mechanism, we conducted a systematic sensitivity analysis across all four reward functions. We randomly sampled 10,000 audio examples from the training set, compressed them into VAE latents, applied Gaussian noise perturbations with varying noise levels ($\sigma \in \{0.01, 0.05, 0.1, 0.5\}$), decoded back to audio, and evaluated the change in reward scores relative to the original unperturbed samples.
>
> The mean ± std of the reward distributions, representing the rough score scale across all samples, are as follows:
>
> *   **Semantic Reward:** 0.180 ± 0.050
>
> *   **Aesthetic Reward:** 0.293 ± 0.088
>
> *   **Spatial Reward:** 0.238 ± 0.194
>
> *   **Temporal Reward:** 0.339 ± 0.170
>
>
> Based on this, the Lipschitz constant K is estimated as:
>
> $K \approx \max_{i} \frac{|R(\mathbf{x}_i + \delta) - R(\mathbf{x}_i)|}{\|\delta\|_2}$
>
> Our empirical results across all 10,000 samples and all noise perturbation levels demonstrate that all four reward functions exhibit approximately Lipschitz continuous behavior:
>
> *   **Semantic Reward:** $K \approx 0.104$ (low sensitivity)
>
> *   **Aesthetic Reward:** $K \approx 0.035$ (very low sensitivity)
>
> *   **Spatial Reward:** $K \approx 0.049$ (very low sensitivity)
>
> *   **Temporal Reward:** $K \approx 1.351$ (moderate sensitivity, yet bounded)
>
>
> These bounded and very-low-to-moderate Lipschitz constants confirm numerical stability across all reward estimations, ensuring error propagation remains controlled during GRPO optimization. All four reward functions demonstrate good robustness to input perturbations.
>
> ---
> > **Q4:** Is the in-window policy fixed as an isotropic Gaussian? If so, did you try diagonal/low-rank/mixture variants?
>
> **Yes, our in-window policy is modeled as an isotropic Gaussian**, i.e., $\mathcal{N}(0, \sigma^2 I)$. We adopt this form to **stay** **strictly consistent with the theoretical SDE-ODE equivalence (Equation 11) inherent to the standard diffusion formulation**. This ensures **the mathematical validity of substituting ODE steps with SDE steps without shifting the marginal distribution**.
>
> While we agree that exploring structured noise (e.g., diagonal or low-rank variants) is a promising avenue for future efficiency gains, our empirical results demonstrate that **the standard isotropic formulation is sufficient to achieve SOTA performance and stable convergence** **in our current framework**.
>
> ---
> > **Q5:** On the VGGSound dataset, how many test videos did you evaluate (exact count for your test split)?
>
> We evaluated our model on **15,060 test videos** from the VGGSound test set, following the same evaluation protocol as ThinkSound[3] by removing silent audios and static videos to ensure fair comparison.

---

> ### Author Response · Authors · 2025-11-21
> **Response to Reviewer 2VpK [3/3]**
>
> > **Ethical Considerations**
>
> We sincerely thank the Reviewer for this invaluable reminder. We fully acknowledge that **ethical considerations are absolutely essential for responsible research**. In the original submission, we have mentioned potential cultural biases regarding what constitutes "good sound" in the Ethical Considerations section. However, our original wording in the "Bias and Representation Issues" section only mentioned the potential issue without detailing the bias mitigations that we had implemented in our original work. We have now augmented Appendix H (Ethical Considerations) with a more comprehensive discussion that explicitly documents the **significant practical mitigations** that we conducted in our original work to ameliorate these potential biases (see Appendix H for details), as follows:
>
> **1. "Aesthetics" is Grounded in Technical Quality, Not Purely Subjective Preference:**
>
> A crucial aspect of our framework is how we define and operationalize "aesthetics." This is not a vague, culturally-loaded preference. As stated in Line 210 of our paper, our **Aesthetic CoT** focuses on tangible audio quality aspects like **"naturalness and fidelity".** This is reinforced by our choice of the reward model. As detailed in Appendix F.1, the Meta Audiobox Aesthetics model[4] provides multi-faceted Aesthetics scores. While it includes a subjective "Content Enjoyment (CE)" score, our optimization also heavily relies on its **"Production Quality (PQ)"** metric. PQ explicitly measures **technical aspects** like **clarity, fidelity, dynamics, and frequency response**, rather than subjective tastes. By optimizing for high fidelity and technical clarity, we primarily push the model towards realism and professional production standards, which are far more universal and less culturally specific than artistic or stylistic preferences. This anchors our "aesthetic" goal to a more objective standard.
>
> **2. Scope is Focused on Non-Human Sounds, Reducing Direct Demographic Bias:**
>
> The scope of our work is mainly focused on **sound effects and instrumental music, deliberately excluding human speech and singing**. This significantly reduces the risk of perpetuating some of the most harmful forms of representational bias related to accents, dialects, language, or vocal characteristics tied to specific demographic groups (e.g., gender, ethnicity). In future work, when incorporating human speech and singing, we will explore effective methods to mitigate these biases.
>
> **3. The Subjective Component of the Reward is Based on a Diverse Rater Pool:**
>
> For the more subjective components of the Aesthetic Reward (such as Content Enjoyment), we deliberately chose the Meta Audiobox Aesthetics model. As emphasized in its technical report[4], this model was trained on ratings from **158 diverse raters from the general public**, explicitly aiming to capture a broad spectrum of human judgments. By leveraging a reward signal that already aggregates diverse opinions, we actively avoid encoding a single, narrow cultural viewpoint for the more subjective aspects of sound quality.
>
> While we do not claim to have eliminated all bias, we believe our specific methodological choices—from the scope of our task to the definition and measurement of "aesthetics"—serve as significant, practical mitigation steps.
>
> ---
>
> ## References
>
> [1] Song, Yang, et al. "Score-based generative modeling through stochastic differential equations." ICLR 2021.
>
> [2] Øksendal, Bernt. "Stochastic differential equations." Stochastic differential equations: an introduction with applications. Berlin, Heidelberg: Springer Berlin Heidelberg, 2003. 65-84.
>
> [3] Liu, Huadai, et al. "Thinksound: Chain-of-thought reasoning in multimodal large language models for audio generation and editing." arXiv preprint arXiv:2506.21448 (2025).
>
> [4] Tjandra A, Wu Y C, Guo B, et al. Meta audiobox aesthetics: Unified automatic quality assessment for speech, music, and sound[J]. arXiv preprint arXiv:2502.05139, 2025.

---

> ### Author Response · Authors · 2025-11-27
> **Gentle Reminder Regarding Our Response**
>
> We sincerely thank the reviewer for your positive feedback and insightful comments, which have been very helpful in strengthening our work.
>
> We are writing to respectfully invite you to review our rebuttal. To further solidify the validity of our work and address your concerns, we have provided detailed theoretical proofs and supplementary experiments, particularly in our responses to Q1-Q3.
>
> We deeply appreciate the time you have dedicated to our paper, and we hope our response helps to reinforce your positive evaluation.

---

### Author Response · Authors · 2025-11-21
**Global Response to All Reviewers and ACs**

We sincerely thank all the reviewers and the ACs for your time and expertise. We appreciate the positive remarks from all three reviewers highlighting our contributions, including `clear problem definition of objective entanglement and addressing it by decomposing reasoning into four CoT axes with aligned, axis-wise rewards under reinforcement learning` (Reviewer 2VpK) and `the conceptual soundness and the strong motivation of the four decomposition dimensions` (Reviewer BAuq), `the Fast-GRPO innovation for efficient training` (Reviewer 2VpK), `the comprehensive experimental validation demonstrating the effectiveness of our multi-dimensional CoT framework` (Reviewer oFRj), including `the high audio quality and SOTA performance with faster inference and strong robustness` (Reviewer 2VpK, BAuq), and `the comprehensive ablations supporting the effectiveness of the proposed framework` (Reviewer oFRj), as well as recognition of `the value and high quality of the AudioCanvas benchmark` (Reviewer BAuq).

We are grateful for all the reviewers' constructive feedback. Below, we address all the questions and concerns raised by multiple reviewers. We have also carefully addressed all the remaining questions from each reviewer in our respective, point-by-point response to each reviewer. In addition, we have conducted a comprehensive revision of the manuscript. Please refer to our comment on **"The List of Revisions Made to the Manuscript"**.

---

**Q1: GPU Resource Requirements and Accessibility**

Our PrismAudio is designed for practical deployment with efficient inference. On NVIDIA A800 GPUs with batchsize=1, inference consumes approximately **5.6GB of VRAM**, making it accessible for consumer-grade GPUs. While the complete training pipeline requires 16-24 NVIDIA 80GB A800 over 2-3 weeks, which is typical for foundation models, our design ensures democratized access through efficient inference and open weights. Specifically, the training pipeline consists of: **(1)** VAE fine-tuning (optional, 24 A800, ~5 days), **(2)** Main model training (16 A800, ~3 days), **(3)** Fast-GRPO post-training (8 A800, ~5 days), and **(4)** VideoLLaMA2 fine-tuning (8 A800, ~2 days). **Importantly, VAE fine-tuning is optional** and can be skipped with a performance trade-off (see Table 10); Fast-GRPO **reduces training time by 1.6×** compared to full SDE-GRPO. We will release all pre-trained model checkpoints for direct inference use.

---

**Q2: Reproducibility and Open-Source Commitment**

We acknowledge concerns about the complexity of our multi-reward training pipeline. Our multi-reward design is essential for addressing the objective entanglement problem, and we have taken comprehensive measures to ensure reproducibility and minimize dependency issues: Our framework uses four reward signals (as detailed in Section 3.3.1), **all of which are open-source with well-documented inference pipelines**. While configuring dependencies for four different reward model libraries is non-trivial, **we have verified that all four libraries can be successfully configured in a single environment and function correctly for inference**.

To ensure full reproducibility, we are committed to open-sourcing: **(1)** complete, well-documented code for all components, **(2)** pre-trained model checkpoints, **(3)** the AudioCanvas benchmark and both versions of CoT captions (**generated from sounding videos and silent videos**) to accommodate different deployment scenarios, **(4)** Docker containers with all dependencies pre-configured, **(5)** automated setup scripts, and **(6)** comprehensive documentation, upon acceptance of the paper.

Detailed reproducibility measures are documented in the **Reproducibility Statement** Section of our revised manuscript.

---

**Q3: Project Page and Qualitative Demonstrations**

Our project page (shown in the last sentence of Abstract) provides extensive qualitative evidence demonstrating PrismAudio's superior performance across diverse scenarios. The project page includes side-by-side comparisons across in-domain (VGGSound) and out-of-domain (AudioCanvas) evaluations, single-event and multi-event scenarios, and audio generation for Sora2 and Veo3-generated videos, providing comprehensive qualitative validation of the effectiveness of our framework.

As shown on the project page demonstrations, **PrismAudio resolves hallucination issues** (spurious voices, background music) observed in the previous SOTA ThinkSound and MMAudio. As demonstrated in the _"Comparison with Baselines in Multi-Event Scenarios on the AudioCanvas Benchmark"_ section, **PrismAudio generates clean, contextually appropriate audio for complex scenarios** like "Chopping, Thudding" and "Machining, Tapping" while baseline methods produce unexpected artifacts. Additionally, music examples demonstrate **PrismAudio's superior audio quality with enhanced fidelity, naturalness, and audio-visual synchronization compared to previous SOTA methods**.

---

### Author Response · Authors · 2025-11-21
**The List of Revisions Made to the Manuscript**

To fully address all the questions and concerns from all the reviewers, we have made the following major revisions in the updated manuscript (**all updated text is highlighted in blue**):

1.  **More Ablation Studies and Empirical Evidence (Section 4, Appendix E):** Added multiple experimental analyses: (1) Empirical validation of ODE-SDE distribution equivalence (Appendix E.6, Figure 5), tracking ODE and SDE distributions throughout training; (2) More ablation study of the four-dimensional framework (Appendix E.7, Table 13), validating the necessity of all four dimensions and the superiority of the full configuration over reduced configurations; (3) VAE fine-tuning analysis (Appendix E.3, Table 10), demonstrating that fine-tuning provides improved reconstruction abilities, and offering researchers the flexibility to use our pre-trained VAE for better generation quality; (4) Additional AudioCanvas experiments using CoTs generated from silent videos (Table 2).

2.  **Enhanced Ethical Considerations Discussion (Appendix H):** Expanded the "Bias and Representation Issues" section with a detailed presentation of our three practical mitigations of the biases: (1) grounding aesthetics in technical quality metrics, (2) focusing on non-human sounds to reduce demographic bias, and (3) using diverse rater pools for subjective components.

3.  **Complete Implementation Details and Reproducibility (Appendix D):** Added comprehensive documentation including: (1) Clarified CoT generation process for different evaluation scenarios (AudioCanvas vs. VGGSound); (2) Complete prompts for Gemini 2.5 Pro and training details of VideoLLaMA2 (Appendix D.2, D.3); (3) Reproducibility statement with code release plans, Docker containers, and dependency management; (4) Detailed GPU resource requirements for inference (5,618 MiB VRAM) and the training pipeline (16-24 A800 over 2-3 weeks) with clear indication of optional components.


We hope that this revised manuscript addresses all of the questions and concerns raised by the reviewers, and if not, we would be happy to engage in further discussions and revisions of the paper to best present our work.

---

### Meta-Review · Area_Chair_td9q · 2026-01-05

**Summary:**

This paper introduces PrismAudio, a video-to-audio generation framework using four-dimensional reasoning and reinforcement learning. It achieves top results by solving objective conflicts and using an efficient training method named Fast-GRPO. However, the training process is complex and relies on many different reward models. Reviewers also worried about the novelty of the sampling method and potential train-test gaps. Before the rebuttal, there were doubts about whether the model required audio to generate its reasoning paths. The authors then provided new tests showing the model performs well using only silent videos. The paper is recommended for acceptance, and authors should further simplify the implementation for the final version.

**Reviewer Concerns:**

The authors successfully addressed concerns about the reasoning process for silent videos and the need for four distinct rewards. They also provided proofs for the stability of their training algorithm. The high computational cost and complexity of the pipeline remain as minor outstanding concerns.

**Reviewer Scores:**

Reviewer 2VpK likely would have kept a positive score given the strong theoretical and stability clarifications. Reviewer BAuq would have maintained their high score as the new trade-off analysis confirmed the framework's value. Reviewer oFRj explicitly raised their score because the new experiments proved the model works for real-world silent videos.

---

### Decision · Program_Chairs · 2026-01-26

Accept (Poster)